# PROJECTED NEURAL ADDITIVE MODELS AS UNIVERSAL APPROXIMATORS

## ABSTRACT

This article proves that any continuous multi-variable function can be approximated arbitrarily close by a linear combination of single-variable functions of the inputs in a projected space. Using a set of independent neural networks to parameterize these feature functions of the projected inputs, we introduce their linear combination as the projected neural additive model (PNAM): an extension of the neural additive model (NAM) (cf. Agarwal et al. (2021)) that now enables universal approximation. While the couplings of the input variables bestow the PNAM with the universal approximation property, they could diminish the interpretability intrinsic to the NAM. As such, we propose regularization and post hoc techniques to promote sparse solutions and enhance the interpretability of the PNAM. The single-variable characteristic of the bases also allows us to convert them into symbolic equations and dramatically reduces the number of required parameters. We provide results from numerical experiments on invariants in knot theory, phase field fracture mechanics, and the MNIST benchmark to illustrate the expressivity and interpretability of the PNAM.[1]

## 1 INTRODUCTION

While deep neural networks have become popular for a multitude of tasks due to their expressivity, they come at the cost of interpretability (Murdoch et al., 2019). Of the myriad methods introduced to address this issue, Agarwal et al. (2021) propose an alternative architecture, coined the neural additive model (NAM), by altering the connectivity of the network such that it becomes a linear combination of single-variable functions of the input variables, parameterized by independent multi-layer perceptrons (MLPs). Although the values of these functions can provide a degree of interpretability, the linear nature of the NAM in turn limits its expressivity. As a result, Phan et al. (2025) propose the use of a learnable linear transformation before passing the inputs to the NAM (see Fig. 1), which we refer to as the projected neural additive model (PNAM), to enhance the expressivity of the model.

**Contribution.** In this work, we prove that the PNAM is a universal approximator, elucidating its ability to approximate any continuous function on a closed and bounded domain. In particular, we first prove the polynomial reproducing property of the PNAM for an arbitrary number of variables and orders using mathematical induction; we then employ the Stone–Weierstrass theorem (Stone, 1937; 1948; Cotter, 1990) to establish its universal approximation property. The linear transformation enables the PNAM to capture complex couplings that the NAM and standard generalized additive models (GAMs) cannot, while also reducing the number of feature functions for high-dimensional problems with numerous inputs (Hastie & Tibshirani, 1986; Radenovic et al., 2022).

To rectify the reduction in interpretability that the linear transformation may induce, we introduce regularization techniques that (i) permit us to rank the importance of each input feature, (ii) penalize unnecessary couplings between the inputs, and (iii) promote sparse solutions. While the NAM provides local comprehension by highlighting how the variables affect the predictions at every point, the PNAM offers global comprehension by identifying which variables are crucial for the overall predictive accuracy (Molnar, 2019; Rudin et al., 2022). We further leverage the modularity of the PNAM to prune nonessential parameters and provide an option to convert the single-variable bases into symbolic equations. Applying the resultant models for multi-output predictions, we demonstrate, in

---

[1]We will open-source our code after the double-blind review process.

three numerical experiments, the various utilities of the PNAM, which allow users to dictate their desired degrees of accuracy and sparsity.

**Related work.** The PNAM is one of many architecture-based models introduced to enhance the interpretability of deep neural networks without compromising their expressivity. Other neural network models include, but are not limited to, Kolmogorov–Arnold networks (KANs) (Liu et al., 2024), deep polynomial neural networks (Chrysos et al., 2022), and graph neural networks with inductive biases (Cranmer et al., 2020). In addition, related sparsification, pruning, and post hoc methods include the SINDy algorithm (Brunton et al., 2016), structural pruning (Fang et al., 2023), and gradient-based attribution (Sundararajan et al., 2017). We emphasize that although we have equipped the PNAM with the ability to produce mathematical expressions, symbolic regression (SR) is primarily employed for pruning to reduce the number of parameters and to gain insight into the interactions of the input variables. Similar to the KAN, the PNAM is not intended for recovering physical laws with precise functional forms, as its additive nature prevents the PNAM from compactly approximating operators like division. Moreover, the number of feature functions is potentially large. As a result, we reserve such tasks for proven SR algorithms using reinforcement learning (Petersen et al., 2019), physics-inspired strategies (Udrescu & Tegmark, 2020), and genetic programming (Cranmer, 2023).

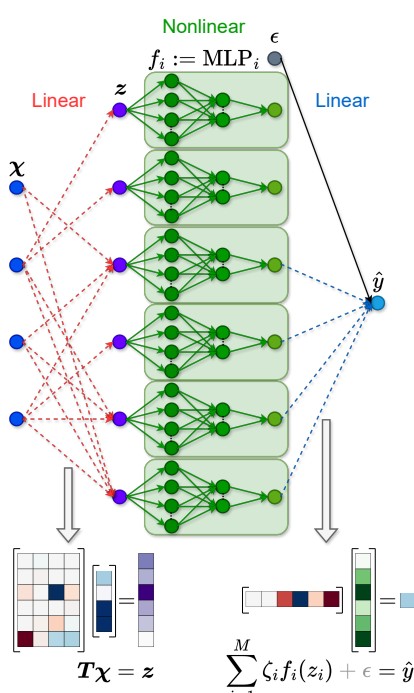

Figure 1: Architecture of the PNAM. The output $\widehat{y}$ is predicted via a linear combination of single-variable functions, parameterized by independent MLPs, which further are functions of linear combinations of the inputs $\boldsymbol{\chi}$. The transformation $\boldsymbol{T}$ and scaling coefficients $\boldsymbol{\zeta}$ (both denoted with dashed lines) can be optimized to yield sparser (and more interpretable) solutions.

## 2 PROJECTED NEURAL ADDITIVE MODELS

### 2.1 CONSTRUCTION

Given a training data set $\{\boldsymbol{\mathcal{X}}, \boldsymbol{y}\} = \{(\chi_j, y_j)\}_{j=1}^{D}$, where $\boldsymbol{\chi} = \{\chi_i\}_{i=1}^{N}$ is an input point, with $N$ denoting the number of independent variables, $y$ is the corresponding scalar output label, and $D$ is the number of input-output pairs, the goal of supervised learning is to construct a function $\mathcal{F}$ that maps every input point to output label, that is, $y = \mathcal{F}(\boldsymbol{\chi})$: $\mathbb{R}^N \to \mathbb{R}$. Here, we hypothesize that the multi-variable function $\mathcal{F}$ can be approximated by a linear combination (or weighted sum) of single-variable functions $\{f_i\}_{i=1}^{M}$, with $M$ denoting the number of feature functions, of the inputs in a projected space to produce the dependent variable:

$$\widehat{y} = \sum_{i=1}^{M} \zeta_i f_i \left( \sum_{j=1}^{N} T_{ij} \chi_j \right) + \epsilon = \sum_{i=1}^{M} g_i(z_i) + \epsilon, \tag{1}$$

where $\widehat{y}$ is a parameterization of $y$, and $\epsilon$ is an error term introduced to represent noise in the data. The projected variables $\boldsymbol{z}$ in Eq. 1 result from a linear transformation $\boldsymbol{T}$ of $\boldsymbol{\chi}$, that is, $\boldsymbol{z} = \boldsymbol{T}\boldsymbol{\chi}$: $\mathbb{R}^N \to \mathbb{R}^M$. Moreover, each single-variable function $f_i$: $\mathbb{R} \to \mathbb{R}$ and its corresponding scaling coefficient $\zeta_i$ are represented by the function $g_i$: $\mathbb{R} \to \mathbb{R}$ for compactness.

The feature functions $\{f_i\}$ in Eq. 1 can be constructed using polynomials or neural networks. Here, we parameterize $y$ as a linear combination of MLPs:

$$\widehat{y} = \sum_{i=1}^{M} \zeta_i \mathrm{MLP}_i \left( \sum_{j=1}^{N} T_{ij} \chi_j; \boldsymbol{W}_i^{(1)}, \dots, \boldsymbol{W}_i^{(L)}, \boldsymbol{s}_i^{(1)}, \dots, \boldsymbol{s}_i^{(L-1)} \right) + \epsilon, \tag{2}$$

where each $f_i$ is an MLP with $L$ layers, learnable weights $\left\{ \boldsymbol{W}_i^{(1)}, \ldots, \boldsymbol{W}_i^{(L)} \right\}$ and biases $\left\{ \boldsymbol{s}_i^{(1)}, \ldots, \boldsymbol{s}_i^{(L-1)} \right\}$, and element-wise activation function $a$:

$$
\begin{aligned}
\mathrm{MLP}_i &\left( z_i; \boldsymbol{W}_i^{(1)}, \ldots, \boldsymbol{W}_i^{(L)}, \boldsymbol{s}_i^{(1)}, \ldots, \boldsymbol{s}_i^{(L-1)} \right) = \\
&\boldsymbol{W}_i^{(L)} a \left( \boldsymbol{W}_i^{(L-1)} a \left( \ldots a \left( \boldsymbol{W}_i^{(1)} z_i + \boldsymbol{s}_i^{(1)} \right) \ldots \right) + \boldsymbol{s}_i^{(L-1)} \right).
\end{aligned}
\tag{3}
$$

To enable the model to learn high-frequency functions, a Fourier feature mapping (Tancik et al., 2020; Bahmani et al., 2024) can be leveraged to map a transformed input $z_i$ to

$$
\gamma_i(z_i) = [\cos(2\pi \boldsymbol{B}_i z_i)^{\mathrm{T}}, \sin(2\pi \boldsymbol{B}_i z_i)^{\mathrm{T}}]^{\mathrm{T}},
\tag{4}
$$

where each entry in $\boldsymbol{B}_i$ is sampled from a Gaussian distribution $\mathcal{N}(0, \sigma^2)$ with standard deviation $\sigma$ and is fixed after initialization, before passing it to Eq. 3. Let $\boldsymbol{\zeta} = \{\zeta_i\}_{i=1}^M$; as shown in Fig. 1, the transformation $\boldsymbol{T}$, scaling coefficients $\boldsymbol{\zeta}$, and error term $\epsilon$ in Eq. 2 can be encoded as two additional weight matrices and one bias term, respectively. A simpler version of this construction is first proposed in Phan et al. (2025) and corresponds to the PNAM, introduced (without a formal proof of universal approximation) to overcome the limited expressivity of the NAM (Agarwal et al., 2021).

*Remark.* For the PNAM, each basis $g_i$ in Eq. 1 is now a function of a transformed variable $z_i$, as opposed to the original input $\chi_j$ for the NAM and other GAMs. The linear transformation $\boldsymbol{T}$, leading to universal approximation, enables the PNAM to capture interactions between the inputs, such as $\chi_1 \chi_2$ in Example A.1, that the NAM cannot. Nevertheless, for each data point, examining $\{g_i\}$ no longer tells us how the inputs $\boldsymbol{\chi}$ contribute to the prediction $\widehat{y}$. Instead, we describe in Section 2.5 how one can examine $\boldsymbol{T}$, which is constant for all data points, to determine which inputs are important for predictive accuracy. This and other aspects of interpretability (e.g., dimensionality reduction and feature pruning) are entirely missing from fully connected MLPs.

## 2.2 UNIVERSAL APPROXIMATION

By the Stone–Weierstrass theorem (Stone, 1937; 1948; Cotter, 1990), polynomials are dense in the space of continuous functions, i.e., they can approximate any continuous function on a closed and bounded domain. As such, we can achieve universal approximation by reproducing polynomials. We show in Theorem A.1 that single-variable polynomials of the inputs in a projected space can be employed as the bases to approximate multi-variable polynomials. Instead of single-variable polynomials, one may also use the sum of one-dimensional (1D) neural networks of the projected inputs to achieve universal approximation. Since all such neural networks of single variables enjoy the universal approximation property (Hornik et al., 1989; Leshno et al., 1993; Lu et al., 2017), implying that they can approximate polynomials, we immediately get the universal approximation property of the resulting multi-dimensional neural network. This approximation capability is formally stated in the following theorem.

**Theorem 2.1.** *Let $\mathfrak{G}$ be a set of single-variable polynomials or 1D neural networks. Given any function $\mathcal{F}_1$ in the set of continuous real-valued functions $\mathcal{C}(\mathfrak{D})$ for a compact domain $\mathfrak{D} \subset \mathbb{R}^N$, there exists a linear transformation $\boldsymbol{T} \colon \mathbb{R}^N \to \mathbb{R}^M$ and a set of functions $\{g_i \colon \mathbb{R} \to \mathbb{R}\}_{i=1}^M$ in $\mathfrak{G}$ such that*

$$
\left| \mathcal{F}_1(\boldsymbol{\chi}) - \sum_{i=1}^M g_i \left( \sum_{j=1}^N T_{ij} \chi_j \right) \right| < \epsilon,
\tag{5}
$$

*for all $\boldsymbol{\chi} \in \mathfrak{D}$ and any $\epsilon > 0$.*

*Proof.* By the Stone–Weierstrass theorem (Stone, 1937; 1948; Cotter, 1990), there exists a (multi-variable) polynomial $\mathcal{F}_2$ such that

$$
|\mathcal{F}_1(\boldsymbol{\chi}) - \mathcal{F}_2(\boldsymbol{\chi})| < \epsilon/2.
\tag{6}
$$

Then for $\mathcal{F}_2$, by Theorem A.1, we have

$$\mathcal{F}_2(\boldsymbol{\chi}) = \sum_{i=1}^{M} \widehat{g}_i \left( \sum_{j=1}^{N} T_{ij} \chi_j \right) \tag{7}$$

with single-variable polynomials $\{\widehat{g}_i\}$. If $\mathfrak{G}$ is a set of polynomials, we are done. If $\mathfrak{G}$ represents neural networks, we know each polynomial $\widehat{g}_i$ can be further approximated by a neural network $g_i$ due to the universal approximation theorem of neural networks (Hornik et al., 1989; Leshno et al., 1993; Lu et al., 2017) in 1D. So, we can make

$$\left| \mathcal{F}_2(\boldsymbol{\chi}) - \sum_{i=1}^{M} g_i \left( \sum_{j=1}^{N} T_{ij} \chi_j \right) \right| < \epsilon/2. \tag{8}$$

This result, together with Eqs. 6 and 7, completes the proof. $\qquad\square$

## 2.3 EXTENSION TO MULTIPLE OUTPUTS

Now, let $\boldsymbol{y} = \{\{y_{ij}\}_{j=1}^{D}\}_{i=1}^{K}$, $\boldsymbol{\zeta} = \{\{\zeta_{ij}\}_{j=1}^{M}\}_{i=1}^{K}$, and $\boldsymbol{\epsilon} = \{\epsilon_i\}_{i=1}^{K}$, with $K$ denoting the number of dependent variables. Equation 1 can be extended to multiple outputs $\{\widehat{y}_i\}$ as follows:

$$\widehat{y}_i = \sum_{j=1}^{M} \zeta_{ij} f_{ij} \left( \sum_{k=1}^{N} T_{jk} \chi_k \right) + \epsilon_i = \sum_{j=1}^{M} g_{ij}(z_j) + \epsilon_i, \qquad i = 1, 2, \ldots, K, \quad (\text{no sum over } i) \tag{9}$$

where $f_{ij}$ is the $i^{\text{th}}$ output of the $j^{\text{th}}$ vector-valued function (Xu et al., 2023). To show that Eq. 9 is a natural extension of Eq. 1, consider the extreme case where the bases $\{g_{ij}\}$ share no common inputs $\{z_j\}$ across $\{\widehat{y}_i\}$. In that case, $M = \sum_{i=1}^{K} m_i$, with $m_i$ denoting the number of dimensions required to approximate each $\widehat{y}_i$. Equivalently, for each index $i$, the number of indices $j$ for which $\zeta_{ij}$ and $f_{ij}$ are nonzero is $m_i$, yielding orthogonal bases $\{g_{ij}\}$. Along the lines of the universal approximation theorem associated with fully connected neural networks, $M$ can be chosen to be arbitrarily large to approximate any set of continuous functions in theory.

## 2.4 LEARNING PROBLEM AND CONSTRAINTS FOR SPARSITY

Let $\boldsymbol{\Theta} = \{\boldsymbol{\theta}_i\}_{i=1}^{M} = \left\{ \left( \boldsymbol{W}_i^{(1)}, \ldots, \boldsymbol{W}_i^{(L)}, \boldsymbol{s}_i^{(1)}, \ldots, \boldsymbol{s}_i^{(L-1)} \right) \right\}_{i=1}^{M}$ denote all learnable parameters of the MLPs. We optimize the learnable parameters of the PNAM by minimizing the following loss function (in parentheses) for $D$ training samples:

$$\begin{aligned}
\boldsymbol{T}^*, \boldsymbol{\Theta}^*, \boldsymbol{\zeta}^*, \boldsymbol{\epsilon}^* = \arg\min_{\boldsymbol{T}, \boldsymbol{\Theta}, \boldsymbol{\zeta}, \boldsymbol{\epsilon}} \bigg( & \mathcal{L}(\boldsymbol{y}, \widehat{\boldsymbol{y}}(\boldsymbol{\mathcal{X}}; \boldsymbol{T}, \boldsymbol{\Theta}, \boldsymbol{\zeta}, \boldsymbol{\epsilon})) + \mathcal{L}_{\text{P}} \\
& + \frac{1}{D}(w_1 \ell_1 + w_2 \ell_2) + \frac{1}{M}(w_3 \ell_3 + w_4 \ell_4 + w_5 \ell_5) \bigg),
\end{aligned} \tag{10}$$

where

$$\mathcal{L} = -\frac{1}{D} \sum_{i=1}^{K} \sum_{j=1}^{D} y_{ij} \log \frac{\exp(\widehat{y}_i(\boldsymbol{\chi}_j; \boldsymbol{T}, \boldsymbol{\Theta}, \boldsymbol{\zeta}, \boldsymbol{\epsilon}))}{\sum_{k=1}^{K} \exp(\widehat{y}_k(\boldsymbol{\chi}_j; \boldsymbol{T}, \boldsymbol{\Theta}, \boldsymbol{\zeta}, \boldsymbol{\epsilon}))}$$

is the cross-entropy loss for classification, or

$$\mathcal{L} = \frac{1}{KD} \sum_{i=1}^{K} \sum_{j=1}^{D} (y_{ij} - \widehat{y}_i(\boldsymbol{\chi}_j; \boldsymbol{T}, \boldsymbol{\Theta}, \boldsymbol{\zeta}, \boldsymbol{\epsilon}))^2$$

is the mean squared error (MSE) for regression. Moreover, $\mathcal{L}_{\text{P}}$ can be employed to impose any additional physical constraints that restrict the space of admissible solutions (Czarnecki et al., 2017; Raissi et al., 2019; Bastek et al., 2024).

To prevent overfitting and produce interpretable solutions, we use the following two constraints:

$$\ell_1 = ||\boldsymbol{\Theta}||_2, \qquad \ell_2 = \left|\left|\{\{\{g_{ij}(\chi_k; \boldsymbol{T}, \boldsymbol{\Theta}, \boldsymbol{\zeta})\}_{k=1}^D\}_{j=1}^M\}_{i=1}^K\right|\right|_2, \qquad (11)$$

where $\ell_1$ is the usual $L_2$ regularization of the weights and biases (Krogh & Hertz, 1991),[2] and $\ell_2$ discourages $\{g_{ij}\}$ from taking on large values (Agarwal et al., 2021). Considering that the inputs and outputs are often scaled to small values in machine learning problems, the inclination for $\{g_{ij}\}$ to be small could be leveraged to determine the relative importance of the inputs. As we will exemplify later, if any element in the linear transformation $\boldsymbol{T}$ is relatively large, its corresponding input feature is more important than the others.

To further promote sparsity, we first define the singular value decomposition of $\boldsymbol{T}$ as follows:

$$\boldsymbol{T} = \boldsymbol{Q}_1 \boldsymbol{\Sigma} \boldsymbol{Q}_2^{\mathrm{T}},$$

where $\boldsymbol{Q}_1$ and $\boldsymbol{Q}_2$ are orthonormal matrices of dimensions $M \times M$ and $N \times N$, respectively, and $\boldsymbol{\Sigma}$ is an $M \times N$ diagonal matrix containing the $\min(M, N)$ singular values of $\boldsymbol{T}$. Based on the metric

$$\Phi(\boldsymbol{R}_1, \boldsymbol{R}_2) = ||\boldsymbol{I} - \boldsymbol{R}_1 \boldsymbol{R}_2^{\mathrm{T}}||_F = \sqrt{2(3 - \mathrm{tr}(\boldsymbol{R}_1 \boldsymbol{R}_2^{\mathrm{T}}))}, \qquad \boldsymbol{R}_1, \boldsymbol{R}_2 \in SO(3),$$

described in Huynh (2009) for measuring the distance between two 3D rotations, we leverage

$$\ell_3 = \Phi(\boldsymbol{I}, \boldsymbol{Q}_1) + \Phi(\boldsymbol{I}, \boldsymbol{Q}_2) = \sqrt{2(M + N - (\mathrm{tr}\boldsymbol{Q}_1 + \mathrm{tr}\boldsymbol{Q}_2))} \qquad (12)$$

to penalize unnecessary couplings between the inputs. In addition,

$$\ell_4 = ||\boldsymbol{T}||_1, \qquad \ell_5 = ||\boldsymbol{\zeta}||_1 \qquad (13)$$

are employed to encourage nonessential coefficients in $\boldsymbol{T}$ and $\boldsymbol{\zeta}$ to go to zero (Tibshirani, 1996; Xu et al., 2023; Bahmani et al., 2024).

## 2.5 POST-PROCESSING AND SYMBOLIC REGRESSION

Upon successful training of the PNAM, post hoc analysis can be performed to further prune the model and enhance interpretability (Murdoch et al., 2019; Cheng et al., 2024). Due to the modularity of the PNAM, we propose three techniques to reduce the number of optimized parameters.

The first technique relies on the successful incorporation of the regularization constraint $\ell_2$ in Eq. 11. Suppose the functions $\{g_{ij}\}$ are indeed small. In that case, we can examine the column-wise mean of the absolute values of the coefficients in the linear transformation (that is, $\frac{1}{M} \sum_{j=1}^M |T_{jk}|$), dubbed the mean absolute coefficients, to rank the importance of each input feature. Upon which, one may choose to keep only the top $n \leq N$ input features and zero out the columns of $\boldsymbol{T}$ associated with the $(N - n)$ less important features. The second technique entails selecting two hyperparameters $T_0$ and $\zeta_0$ for which entries $T_{jk} < T_0$ and $\zeta_{ij} < \zeta_0$, for $k = 1, 2, \ldots, N$, $j = 1, 2, \ldots, M$, and $i = 1, 2, \ldots, K$, are set to zero.[3]

Finally, the third technique leverages the single-variable characteristic of the bases $\{g_{ij}\}$ to convert them into symbolic equations. Although any SR algorithm, such as DSR (Petersen et al., 2019) or AI Feynman (Udrescu & Tegmark, 2020), may be used to accomplish this task, here, we employ PySR (Cranmer, 2023), which utilizes genetic programming (Holland, 1992; Koza, 1994), for its extensive developer base and ease of use. To convert the $i^{\text{th}}$ nonzero output of the $j^{\text{th}}$ MLP into symbolic form, we employ the following loss function:

$$g_{ij}^* = \arg\min_{g_{ij}} \left( \frac{1}{D} \sum_{k=1}^D \left( (\zeta_{ij} \mathrm{MLP}_{ij}(z_j; \boldsymbol{\theta}_j)|_k - g_{ij}(z_j)|_k)^2 \right. \right.$$

$$\left. \left. + w_6 \left( \zeta_{ij} \frac{\mathrm{dMLP}_{ij}(z_j; \boldsymbol{\theta}_j)}{\mathrm{d}z_j} \bigg|_k - \frac{\mathrm{d}g_{ij}(z_j)}{\mathrm{d}z_j} \bigg|_k \right)^2 \right) \right), \quad \text{(no sum over } i \text{ and } j) \qquad (14)$$

---

[2]We opt for $|| \cdot ||_2$ as opposed to $|| \cdot ||_2^2$ so that the (expanded) loss terms in Eq. 10 for regularization constraints $\ell_1, \ell_2, \ldots, \ell_5$ are similar in magnitude. Thus, weighting coefficients $w_1, w_2, \ldots, w_5$ can be chosen together, thereby simplifying the space for hyperparameter search.

[3]Either the first, the second, or a combination of both techniques may be used. Like $M$, $w_1, w_2, \ldots, w_5$, and any other hyperparameters, the choices of $n$, $T_0$, and $\zeta_0$ depend on the users and their desired degrees of accuracy and sparsity.

where $\{g_{ij}\}$ are mathematical expressions of single variables, and the weighting coefficient $w_6$ may be used to control the derivatives of the discovered functions. The sum of $\{g_{ij}\}$ over $j$ (with $\{\epsilon_i\}$) then yields the outputs $\{\widehat{y}_i\}$ in Eq. 9. These post-processing steps can potentially reduce the tens of thousands of parameters of the PNAM to tens or hundreds of parameters, while alleviating the NP-hardness of multi-variable SR (Petersen et al., 2019; Virgolin & Pissis, 2022) and retaining the accuracy of deep neural networks.

## 3 NUMERICAL EXPERIMENTS

In the following experiments, we illustrate the expressivity and interpretability of the PNAM, afforded by the linear transformation and post hoc analysis. The first experiment leverages an extensive data set of mathematical knots from Davies et al. (2021) for (i) multi-label classification and (ii) single-task regression. The second experiment employs limited data from a phase field simulation of fracture propagation in Clayton et al. (2023) for multi-task regression, leveraging the additional physical constraint term $\mathcal{L}_P$ in Eq. 10 and derivative information via the weighting coefficient $w_6$ in Eq. 14. If not specified, $\mathcal{L}_P$ is not used and $w_6$ is set to zero in the experiment. One auxiliary experiment is presented in Appendix A.5, which uses the MNIST data set (LeCun et al., 1998) to demonstrate the dimensionality reduction capability and visualize the input pruning mechanism of the PNAM for a high-dimensional image classification problem.

For all experiments (excluding the one using the MNIST data set), we hold out 20% of the data for testing; the remaining 80% undergo a training-validation split of 80 and 20%, respectively. The PNAM is implemented using the PyTorch deep learning library (Paszke et al., 2019) and SiLU[4] (Hendrycks & Gimpel, 2016; Elfwing et al., 2018) as the activation function $a$ in Eq. 3. For the projection dimension $M$, we start with $M = 8$ or a square projection (whichever is smaller) in every experiment and increase or decrease $M$ as appropriate. Unless otherwise stated, we set weighting coefficients $w_1 = w_2 = w_3 = w_4 = w_5 = 0.01$ for all classification tasks and $w_1 = w_2 = w_3 = w_4 = w_5 = 0.001$ for all regression tasks.[5] We use a batch size of 256 samples and the Adam optimizer (Kingma & Ba, 2014), employing an initial learning rate of 0.001 that decays by a factor of 0.995 after every epoch, to train all neural networks. All models are trained on a single NVIDIA A100-SXM4-40GB GPU. Each basis $g_{ij}$ in Eq. 14 is evolved for one minute using 30 populations of 30 expressions with a maximum complexity of 30; all operators and leaf nodes have a complexity of one. Other relevant hyperparameters and training details are delineated with the results.

### 3.1 BENCHMARKING WITH KNOT THEORY

Established by a team of mostly Google DeepMind (DM) researchers (Davies et al., 2021), the data set of mathematical knots consists of 243,746 samples, each possessing 17 geometric invariants: adjoint torsion degree, torsion degree, short geodesic (real part), short geodesic (imaginary part), injectivity radius, Chern–Simons invariant, cusp volume, longitudinal translation, meridional translation (imaginary part), meridional translation (real part), volume, and six symmetry groups. The goal of this problem is to use the aforementioned invariants to predict the signature of the knots, which can take on one of 14 values that are multiples of 2 from $-12$ to 14. As such, this task can be framed as a classification problem with 14 labels or a single-output regression problem; both options have been explored to benchmark performance.

**Classification.** The first two rows of Table 1 are reproduced from Liu et al. (2024), which compare the performance of the MLP[6] implemented by Davies et al. (2021) against that of the KAN. The next five rows detail our implementation of the MLP, the NAM, and three parameterizations of the

---

[4]We observe that ReLU results in a smaller loss $\mathcal{L}$ in Eq. 10 than SiLU, but the bases $\{g_{ij}\}$ that the PNAM learns are more chaotic/non-smooth and require more parameters/operations to approximate via Eq. 14.

[5]We find that these coefficient values yield a robust trade-off between accuracy and sparsity due to the magnitude of $\mathcal{L}$ relative to those of the regularization constraints in Eq. 10 for scaled variables.

[6]Inspections of the source code (https://github.com/google-deepmind/mathematics_conjectures/blob/main/knot_theory.ipynb) reveal that the training of the MLP is terminated when the validation loss increases (i.e., an early stopping patience of one evaluation is employed), which may have led to underfitting.

Table 1: Performance of different neural network architectures for the multi-label classification problem of predicting the signature of mathematical knots. The first two rows are reproduced from Table 3 of Liu et al. (2024). See Liu et al. (2024) for definitions of $G$ and $k$. In the next five rows, the mean and standard deviation of the test accuracy are computed from 10 runs. In the last row, the first hidden layer of the MLP bases is replaced with a Fourier feature mapping (Eq. 4).

| Method | Architecture | Parameter count | Test acc. |
|--------|-------------|-----------------|-----------|
| DM's MLP | 4 layers: [17, 300, 300, 300, 14] | $3 \times 10^5$ | 78.0% |
| KAN | 2 layers: [17, 1, 14] ($G = 3$, $k = 3$) | $2 \times 10^2$ | 81.6% |
| Our MLP | 3 layers: [17, 64, 32, 14] | $1.2 \times 10^4$ | $95.8 \pm 0.1\%$ |
| NAM | 3 layers: $17 \times [1, 64, 32, 14]$ | $2.1 \times 10^5$ | $92.4 \pm 0.2\%$ |
| PNAM | 3 layers: $17 \times [1, 64, 32, 14]$ | $2.1 \times 10^5$ | $95.0 \pm 0.2\%$ |
| PNAM | 3 layers: $8 \times [1, 64, 32, 14]$ | $9.8 \times 10^4$ | $93.6 \pm 0.4\%$ |
| PNAM | 3 layers: $8 \times [1, 2(32), 32, 14]$ ($\sigma = 1$) | $9.8 \times 10^4$ | $94.3 \pm 0.3\%$ |

Table 2: Ranking of important input features based on their mean absolute coefficients for the PNAM with $M = 8$ in the penultimate row of Table 1. The mean and standard deviation of the test accuracy associated with using only the top $n$ features are computed from their frequency across 10 runs.

| Rank | Input | Symbol | Frequency | Test acc. |
|------|-------|--------|-----------|-----------|
| 1 | Re(meridional translation) | $\chi_{10}$ | 10/10 | $55.1 \pm 3.5\%$ |
| 2 | Longitudinal translation | $\chi_8$ | 10/10 | $74.2 \pm 1.5\%$ |
| 3 | Im(meridional translation) | $\chi_9$ | 7/10 | $78.1 \pm 2.4\%$ |
| | Cusp volume | $\chi_7$ | 1/10 | 80.5% |
| | Im(short geodesic) | $\chi_4$ | 1/10 | 73.5% |
| | Volume | $\chi_{11}$ | 1/10 | 70.5% |

PNAM, all using MLP(s) with two hidden layers of 64 and 32 neurons.[7] We scale all inputs to have zero mean and unit variance and train the models for 50 epochs without early stopping. All five models achieve test accuracy greater than 90%, with the MLP performing the best and the NAM performing the worst. Although the accuracy of the PNAM can be improved by increasing $M$ (e.g., from 8 to 17) or replacing the first hidden layer of the MLP bases with a Fourier feature mapping (e.g., with $\boldsymbol{B} \in \mathbb{R}^{32}$ and $\sigma = 1$), doing so increases the complexity of the learned functions.

For the PNAM with $M = 8$ and without the Fourier feature mapping, we present in Table 2 possible input variables that represent the three most important features by comparing the mean absolute coefficients of the linear transformation $\boldsymbol{T}$. Out of 10 runs, the meridional translation (real part) and longitudinal translation have the largest and second largest mean absolute coefficients, respectively, in all 10 runs, while the meridional translation (imaginary part) has the third largest mean absolute coefficient in seven runs. In addition, Table 2 reveals that keeping only coefficients in $\boldsymbol{T}$ associated with the top $n = 3$ inputs and zeroing out all other coefficients, the PNAM can still achieve a test accuracy of 78.1% (see Fig. A.1 for more information). Our findings are consistent with Fig. 3 of Davies et al. (2021)[8] and Fig. 4.3 of Liu et al. (2024), despite the fact that Davies et al. (2021) employ gradient-based attribution (Sundararajan et al., 2017) and Liu et al. (2024) leverage a specific KAN architecture with a hidden dimension of one to determine the relative importance of the inputs.

**Regression.** Davies et al. (2021) and Liu et al. (2024) then leverage the knowledge they acquire from the classification task to construct mathematical expressions for the signature of the knots, now as a single-output regression problem. Expression A in Table 3 corresponds to the equation handcrafted by Davies et al. (2021), and expressions B to F proceed from post-processing steps of

---

[7]The number of parameters of the MLP, NAM, and PNAM is estimated as $O(LW^2)$, $O(NLW^2)$, and $O(MN + MLW^2)$, respectively, where $W$ is the number of neurons in the widest layer. Compared to an MLP that uses the same $L$ and $W$, the PNAM has approximately $M$ times more parameters. Considering that memory requirements scale linearly with the number of parameters, the PNAM requires $M$ times more memory and is thus slower to train than the MLP.

[8]Note that Davies et al. (2021) swap the naming of the real and imaginary parts of the meridional translation in their code/figure (see footnote 6).

Table 3: Mathematical expressions for the knot data set. Inputs $\chi_7$, $\chi_8$, $\chi_9$, and $\chi_{10}$ are the cusp volume, longitudinal translation, meridional translation (imaginary part), and meridional translation (real part), respectively. Expressions A to F are reproduced from Table 4 of Liu et al. (2024). A factor of $\frac{1}{2}$ is added to expression A for consistency with DeepMind's findings (Davies et al., 2021; 2024). Expression D has missing parentheses, so it cannot be evaluated. A factor of $\frac{1}{2}$ is added to expression E for consistency with expression A. For expressions G and H, bases $\{g_{ij}\}$ in Eq. 14 use addition, subtraction, multiplication, and square as operators; they additionally use exponential, sine, and tangent for expression I. Every constant in the expressions is counted as a parameter. See our code to reproduce these results.

| ID | Expression | PC† | Discovered by | Eval. of test acc. | | Total acc. |
| --- | --- | --- | --- | --- | --- | --- |
| | | | | Reported | Our | |
| A | $\frac{\chi_8\chi_{10}}{2(\chi_{10}^2+\chi_9^2)}$ | 3 | Human (DM) | 83.1% | 74.5% | 73.8% |
| B | $-0.02\sin(4.98\chi_9+0.85)+0.08\lvert 4.02\chi_{10}+6.28\rvert-0.52-0.04e^{-0.88(1-0.45\chi_8)^2}$ | 12 | [3, 1] KAN | 62.6% | 27.0% | 26.8% |
| C | $0.17\tan(-1.51+0.1e^{-1.43(1-0.4\chi_9)^2+0.09e^{-0.06(1-0.21\chi_8)^2}}+1.32e^{-3.18(1-0.43\chi_{10})^2})$ | 17 | [3, 1, 1] KAN | 71.9% | 41.7% | 41.5% |
| D | $-0.09+1.04\exp(-9.59(-0.62\sin(0.61\chi_{10}+7.26))-0.32\tan(0.03\chi_8-6.59)+1-0.11e^{-1.77(0.31-\chi_9)^2}-1.09e^{-7.6(0.65(1-0.01\chi_8)^3}+0.27\arctan(0.53\chi_9-0.6)+0.09+\exp(-2.58(1-0.36\chi_{10})^2))$ | 29 | [3, 2, 1] KAN | 84.0% | – | – |
| E | $\frac{4.76\chi_8\chi_{10}}{2(3.09\chi_9+6.05\chi_{10}^2+3.54\chi_9^2)}$ | 7 | [3, 2, 1] KAN + Padé approx. | 82.8% | 79.3% | 79.3% |
| F | $\frac{2.94-2.92(1-0.10\chi_{10})^2}{0.32(0.18-\chi_{10})^2+5.36(1-0.04\chi_8)^2+0.50}$ | 13 | $\frac{[3,1] \text{ KAN}}{[3,1] \text{ KAN}}$ | 77.8% | 27.0% | 26.8% |
| G | $12.766(0.132(-\chi_{10}+0.035\chi_8+0.157)^2+0.592(-0.23\chi_{10}+0.008\chi_8+1)^2(0.162\chi_{10}-0.006\chi_8+0.076)-1)^4+7.202(0.871\chi_{10}+0.029\chi_8-(0.229\chi_{10}+0.008\chi_8-0.103)(0.229\chi_{10}+0.008\chi_8+0.159(\chi_{10}+0.033\chi_8-0.449)^2+0.625)-0.173)^2-10.643$ | 31 | 8 × [1, 64, 32, 1] PNAM (n = 2) | – | 75.9% | 75.7% |
| H | $26((0.096\chi_{10}-0.002\chi_7+0.004\chi_8-0.237)(0.267\chi_{10}-0.004\chi_7+0.011\chi_8+0.119(\chi_{10}-0.016\chi_7+0.04\chi_8-0.316)^2-1)^2+0.133)(0.734\chi_{10}-0.012\chi_7+0.029\chi_8+0.418)+2.054(0.589\chi_{10}+0.018\chi_7-0.027\chi_8+1)^2-4.056(\chi_{10}+0.031\chi_7-0.046\chi_8+0.195)^2(0.107\chi_{10}+0.003\chi_7-0.005\chi_8-0.064(\chi_{10}+0.031\chi_7-0.046\chi_8+0.087(\chi_{10}+0.031\chi_7-0.046\chi_8-0.165(\chi_{10}+0.031\chi_7-0.046\chi_8+0.195)^2+0.195)^2+0.195)^2+1)^2-0.895$ | 52 | 8 × [1, 64, 32, 1] PNAM (n = 3) | – | 81.2% | 80.9% |
| I | $2.574\chi_{10}+0.078\chi_7-0.13\chi_8+26(0.233(-\chi_{10}+0.016\chi_7-0.04\chi_8-0.597)^2\sin^2(0.367\chi_{10}-0.006\chi_7+0.015\chi_8-0.719)+\sin(0.777\chi_{10}-0.013\chi_7+0.031\chi_8+0.262))(0.168\chi_{10}-0.003\chi_7+0.007\chi_8-0.053)+15.626(-\sin(0.54\chi_{10}+0.017\chi_7-0.025\chi_8+0.105)-0.049)^2(-0.165\chi_{10}-0.005\chi_7+0.008\chi_8-0.645)+0.509$ | 34 | 8 × [1, 64, 32, 1] PNAM (n = 3) | – | 81.4% | 81.0% |

† Parameter count is abbreviated as PC.

KANs trained using only the three most relevant invariants (Liu et al., 2024). Davies et al. (2021) originally report a test accuracy between 70–80% for expression A in their implementation, while Liu et al. (2024) report a test accuracy of 83.1%. Since the accuracy appears to depend on the test set that results from a random data split, we evaluate all expressions on our test set and the entire knot data set (denoted as "Our" and "Total acc." in Table 3), in addition to the values reported by Liu et al. (2024) for expressions A to F.

Here, we demonstrate that the PNAM in Table A.1—trained using all invariants and the same setup described in the classification task but without any knowledge of prior results—can be converted into symbolic equations with merely tens of parameters. We emphasize that expressions G to I in Table 3 are artificial constructs of the PNAM after pruning. Their particular forms are less crucial and would likely change as more analysis becomes available. Instead, what is crucial is the capability of the PNAM to discover pertinent relationships as new data and invariants are introduced.

Of the runs summarized in Table A.2, expressions H and I are obtained from the run depicted in Fig. A.2, with $n = 3$ and the cusp volume[9] as the third most important feature. Although the cusp volume is not explicitly stated in Davies et al. (2021) and Liu et al. (2024) as an invariant relevant for predictive accuracy, the PNAM discovers a potential relationship between the cusp volume and the signature of the knots that could improve accuracy. Furthermore, comparing expressions E and G suggests that the PNAM can achieve accuracy similar to that of the KAN, despite the PNAM using only two invariants without relying on additional assumptions. For additional analyses of how the weighting coefficients in Eq. 10 affect the performance of the PNAM, see Fig. A.5.

## 3.2 PHASE FIELD THEORY FOR FRACTURE OF BRITTLE SOLIDS

In this common solid mechanics problem, we examine the relationship between the expressivity of the PNAM and its projection dimension $M$. The data set simply contains 96 data points,[10] homogenized from a phase field simulation of fracture in boron carbide ($B_4C$) with isotropic elasticity and isotropic fracture energy from Clayton et al. (2023), for quasi-static extension up to peak load. Given the homogenized values of the axial strain, order parameter, and magnitude of the material gradient of the order parameter, the goal of this problem is to predict the average strain energy, phase energy, and axial stress. Considering that the stress is calculated as the derivative of the sum of the strain energy and phase energy with respect to the strain, we frame this task as a two-output regression problem. The MSE is employed as $\mathcal{L}$ in Eq. 10 to predict the strain energy and phase energy; to predict the stress, we use the following form of the constraint term $\mathcal{L}_P$:[11]

$$\mathcal{L}_P = \frac{w_{P1}}{D} \sum_{k=1}^{D} \left( (y'_{1,1} + y'_{2,1})|_k - \frac{\partial(\widehat{y}_1(\boldsymbol{\chi}; \boldsymbol{T}, \boldsymbol{\Theta}, \boldsymbol{\zeta}, \boldsymbol{\epsilon}) + \widehat{y}_2(\boldsymbol{\chi}; \boldsymbol{T}, \boldsymbol{\Theta}, \boldsymbol{\zeta}, \boldsymbol{\epsilon}))}{\partial \chi_1} \bigg|_k \right)^2,$$

where $y'_{1,1}$ and $y'_{2,1}$ are the derivatives of the strain energy and phase energy, respectively, with respect to the strain ($\chi_1$). We set weighting coefficient $w_{P1} = 1$.

All variables are scaled to have zero min and unit max. Table 4 depicts the performance of the MLP, the NAM, and four parameterizations of the PNAM, all using MLP(s) with three hidden layers of 256 neurons and set to train for 5000 epochs with an early stopping patience of 50 epochs. On average, the test MSE of the PNAM decreases as we increase $M$ (cf. Phan et al. (2025)). Nevertheless, a run of the PNAM with $M = 8$ turns out to be the model that achieves the smallest MSE. Therefore, we leverage $w_6 = 1$ in Eq. 14, which ensures accurate predictions of the stress, to convert this neural network model with approximately $10^6$ parameters into symbolic form with roughly 100 parameters. As illustrated in Figs. A.7 and A.8, by employing SiLU as the activation function $a$ in Eq. 3, we are able to approximate all bases $\{g_{ij}\}$ as polynomials of single variables in Table 5. Predictions of the strain energy, phase energy, and stress for the linear combinations of these polynomials are portrayed in Fig. A.9, achieving a test MSE of $2.25 \times 10^{-5}$.

## 4 CONCLUSION

We prove the universal approximation property of the PNAM and demonstrate its superior prediction capability compared to the NAM in three numerical experiments. By increasing the dimension of the linear transformation, the PNAM can achieve performance comparable to or even surpass that of the MLP. Moreover, we leverage the modularity of the PNAM to gain insight into important input features, prune unnecessary parameters, and convert the model into symbolic form. However, as a stand-alone model, the PNAM is not meant to replace the MLP, NAM, or even classical SR. Rather, it serves as an alternative to obtain a better trade-off between expressivity and interpretability— achieving the accuracy of the MLP and retaining a degree of interpretability of the NAM, all while being relatively straightforward to train and optimize. Further studies are required before we can

---

[9]The cusp volume is equivalent to the multiplication of the longitudinal translation and the imaginary part of the meridional translation (see Fig. 4.4(b) of Liu et al. (2024)).

[10]The phase field data set is open-source with our code and can be used to study the benign or catastrophic overfitting of overparameterized neural networks (Mallinar et al., 2022) for a physical system.

[11]Ginzburg–Landau kinetics (Gurtin, 1996), or its quasi-static reduction in the present case, could be added as a constraint if one has access to the loading rate and the Laplacian of the order parameter. See Miehe et al. (2010) for background on phase field fracture mechanics and corresponding numerical models.

Table 4: Performance of different neural network architectures for the multi-output regression problem of predicting the strain energy, phase energy, and stress. The mean and standard deviation of the test MSE for all three variables are computed from 10 runs.

| Method | Architecture | Parameter count | Test MSE |
|---|---|---|---|
| MLP | 4 layers: [3, 256, 256, 256, 2] | $2.6 \times 10^5$ | $(7.05 \pm 2.02) \times 10^{-5}$ |
| NAM | 4 layers: $3 \times [1, 256, 256, 256, 2]$ | $7.9 \times 10^5$ | $(1.12 \pm 0.62) \times 10^{-4}$ |
| PNAM | 4 layers: $3 \times [1, 256, 256, 256, 2]$ | $7.9 \times 10^5$ | $(3.31 \pm 2.11) \times 10^{-4}$ |
| PNAM | 4 layers: $8 \times [1, 256, 256, 256, 2]$ | $2.1 \times 10^6$ | $(1.34 \pm 0.91) \times 10^{-4}$ |
| PNAM | 4 layers: $16 \times [1, 256, 256, 256, 2]$ | $4.2 \times 10^6$ | $(8.01 \pm 3.76) \times 10^{-5}$ |
| PNAM | 4 layers: $32 \times [1, 256, 256, 256, 2]$ | $8.4 \times 10^6$ | $(6.47 \pm 2.71) \times 10^{-5}$ |

Table 5: Mathematical expressions for the phase field data set. Inputs $\chi_1$, $\chi_2$, and $\chi_3$ are the strain, order parameter, and material gradient of the order parameter, respectively. To convert the PNAM with $M = 8$ in Table 4 and $T_0 = \zeta_0 = 0.05$ into symbolic form, bases $\{g_{ij}\}$ in Eq. 14 use addition, subtraction, multiplication, and square as operators. Counting every constant in $\{g_{ij}\}$ and $\{z_j\}$ as a parameter, the symbolic form has 112 parameters and achieves a test MSE of $2.25 \times 10^{-5}$.

| Basis | Expression | Input | Expression |
|---|---|---|---|
| $g_{1,1}$ | $-z_1^3 + 1.524z_1^2 + z_1 + ((-z_1 + (z_1 - 0.028)^2 + 0.75)^2 - 0.102)^2 - 0.269$ | $z_1$ | $203.741\chi_1 + 1.87\chi_2 - 3.048\chi_3$ |
| $g_{2,1}$ | $z_1^2 - (z_1 - 0.597)(z_1 - 0.291)(1.378z_1(0.786z_1 - 1)^4 + 0.525z_1 - 0.074) - 0.093$ | | |
| $g_{1,2}$ | $z_2^3(0.147 - 0.074z_2) + 0.383z_2^2 + 0.119z_2 - 0.119$ | $z_2$ | $235.816\chi_1 - 0.461\chi_2 + 2.333\chi_3$ |
| $g_{2,2}$ | $-10.229z_2(0.003 - 0.002z_2^2) + 0.011$ | | |
| $g_{1,3}$ | $0.967z_3(z_3 - 1.108)(-3.255z_3^3 + z_3 - 1.418) - 2z_3 + 0.07$ | $z_3$ | $-114.828\chi_1 + 1.284\chi_2 + 12.089\chi_3$ |
| $g_{2,3}$ | $1.046z_3 + 2.265(z_3 - 0.403)^2(0.65z_3^3 - z_3^2 + 0.835)^2 - 0.25$ | | |
| $g_{1,4}$ | $-z_4(0.009z_4 + 0.069) + 0.005$ | $z_4$ | $-185.754\chi_1 + 1.866\chi_2 + 2.19\chi_3$ |
| $g_{2,4}$ | $0$ | | |
| $g_{1,5}$ | $0.118z_5(2z_5 + 0.217)(0.033z_5^2(z_5 + 1) - z_5 + 3.239) - 0.116$ | $z_5$ | $190.698\chi_1 + 0.282\chi_2 + 6.271\chi_3$ |
| $g_{2,5}$ | $-z_5(-0.03z_5 + 0.003(0.772z_5(z_5 - 0.466)^2 - z_5 + 0.596)^2 + 0.009) - 0.002$ | | |
| $g_{1,6}$ | $0$ | $z_6$ | $1.503\chi_2$ |
| $g_{2,6}$ | $z_6(0.003z_6^2 + 0.009z_6 - 0.043) + 0.016$ | | |
| $g_{1,7}$ | $z_7(z_7^2(z_7^{10}(z_7 + 0.917)^2 - 0.014) + 0.079z_7 - 0.087) - 0.029$ | $z_7$ | $-119.681\chi_1 + 0.531\chi_2 - 5.929\chi_3$ |
| $g_{2,7}$ | $-0.004z_7^3 + 0.009z_7^2 + 0.024z_7 + 0.007$ | | |
| $g_{1,8}$ | $0.003 - 0.009z_8$ | $z_8$ | $-27.984\chi_1 - 0.769\chi_2 + 6.628\chi_3$ |
| $g_{2,8}$ | $0$ | | |

recommend the PNAM, because of its modularity, as a backbone in graph neural networks or transformers, which may allow us to prune the vast number of parameters of these models and accelerate inference during test time.

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

# A APPENDIX

## A.1 POLYNOMIAL REPRODUCING

Here, we prove the polynomial reproducing property, namely, any multi-variable polynomial $\mathcal{F}$ of $\chi \in \mathbb{R}^N$ can be reproduced by a linear combination of $\{g_i\}$, where $g_i(z_i)$ is a single-variable polynomial of $z_i = \sum_{j=1}^N T_{ij}\chi_j$, for each $i = 1, 2, \ldots, M$. The main proof can be established for arbitrary dimensions with mathematical induction. The base case for this proof by induction is conducted in two dimensions.

First, we present a lemma that verifies the reproducing property for a special case in two dimensions, which can be used to prove the general case.

**Lemma A.1.** *For any monomial $\chi_1^p \chi_2^q$, there exists a linear transformation $\boldsymbol{T}: \mathbb{R}^N \to \mathbb{R}^M$, where $N = 2$ and $M = p + q + 1$, with $T_{i1} = 1$ and $T_{i2} = c_i$, and a set of coefficients $\{\zeta_i\}$ such that*

$$\chi_1^p \chi_2^q = \sum_{i=1}^M \zeta_i \left( \sum_{j=1}^N T_{ij}\chi_j \right)^{M-1} = \sum_{i=1}^M \zeta_i (\chi_1 + c_i \chi_2)^{p+q}. \tag{A.1}$$

*Proof.* Let us expand each term of the summation on the right-hand side of Eq. A.1 and collect terms with the same polynomial orders. We get the following equivalent form:

$$\chi_1^p \chi_2^q = \sum_{i=1}^M \zeta_i (\chi_1 + c_i \chi_2)^{p+q}$$

$$= \sum_{m=0}^{p+q} \sum_{i=1}^M c_i^{p+q-m} \zeta_i \begin{pmatrix} p+q \\ m \end{pmatrix} \chi_1^m \chi_2^{p+q-m}.$$

Comparing the coefficients of the polynomials on both sides, for any choice of $\{c_i\}$, we end up with an $M \times M$ system of linear equations for $\{\zeta_i\}$ as follows:

$$\begin{bmatrix} c_1^{p+q} & c_2^{p+q} & c_3^{p+q} & \cdots & c_M^{p+q} \\ c_1^{p+q-1} & c_2^{p+q-1} & c_3^{p+q-1} & \cdots & c_M^{p+q-1} \\ c_1^{p+q-2} & c_2^{p+q-2} & c_3^{p+q-2} & \cdots & c_M^{p+q-2} \\ \vdots & \vdots & \vdots & \ddots & \vdots \\ 1 & 1 & 1 & \cdots & 1 \end{bmatrix} \begin{bmatrix} \zeta_1 \\ \zeta_2 \\ \zeta_3 \\ \vdots \\ \zeta_M \end{bmatrix} = \delta_{mp} \begin{bmatrix} 1 \\ \frac{1}{p+q} \\ \begin{pmatrix} p+q \\ 2 \end{pmatrix}^{-1} \\ \vdots \\ 1 \end{bmatrix},$$

where $\delta_{mp}$ is the Kronecker delta. Each equation of the linear system can be written compactly as

$$\sum_{i=1}^{M} c_i^{p+q-m} \zeta_i = \delta_{mp} \begin{pmatrix} p+q \\ m \end{pmatrix}^{-1}, \qquad m = 0, 1, \ldots, p+q.$$

To show that we can find $\{c_i, \zeta_i\}$ satisfying the system, we note that the coefficient matrix forms a so-called Vandermonde matrix (Macon & Spitzbart, 1958; Turner, 1966), which is invertible as long as $\{c_i\}$ is taken to be any set of mutually different constants, that is, $c_i \neq c_j$ for $i \neq j$. This completes the proof. $\square$

Even for this simple case, we may observe the nonuniqueness in the construction of the linear transformation. Therefore, it is reasonable to expect that, for specific application problems involving machine learning, one may optimize the transformation to achieve effective representations in a possibly low-dimensional feature space. We give some illustrations in the following example.

**Example A.1.** *We present some instances where the actual value of $M$ could be smaller than $p + q + 1$. In fact, for $p = q = 1$, we have*

$$\chi_1 \chi_2 = \left( \chi_1 + \frac{1}{4}\chi_2 \right)^2 - \left( \chi_1 - \frac{1}{4}\chi_2 \right)^2,$$

*corresponding to $M = 2$. For the trivial case of $p = 0$ or $q = 0$, we may simply use the identity transformation with $M = N = 1$. There are also examples that have a smaller value of $M$ than the input dimension $N = 2$, such as $\chi_1^2 + 2\chi_1\chi_2 + \chi_2^2 = z_1^2$, where $z_1 = \chi_1 + \chi_2$, for which $M = 1 < 2 = N$. This again illustrates the effect of possible dimensionality reduction via a suitable transformation.*

Next, we consider the extension of Lemma A.1. Since a product of polynomials of single variables $f_1(\chi_1)$ and $f_2(\chi_2)$ can be written as linear combinations of monomials $\{\chi_1^p \chi_2^q\}$, Lemma A.1 can be easily extended to a slightly more general form.

**Lemma A.2.** *For any polynomial of $(\chi_1, \chi_2) \in \mathbb{R}^2$ having the product form $f_1(\chi_1)f_2(\chi_2)$, there exists a linear transformation $\boldsymbol{T}$: $\mathbb{R}^2 \to \mathbb{R}^M$, where $M = \deg(f_1) + \deg(f_2) + 1$, and a set of single-variable polynomials $\{g_i: \mathbb{R} \to \mathbb{R}\}_{i=1}^{M}$ such that*

$$f_1(\chi_1)f_2(\chi_2) = \sum_{i=1}^{M} g_i(T_{i1}\chi_1 + T_{i2}\chi_2), \tag{A.2}$$

where $\{\deg(f_i)\}_{i=1}^{2}$ denote the degrees of the single-variable polynomials $f_1$ and $f_2$, the highest exponents of the variables $\chi_1$ and $\chi_2$ in $f_1$ and $f_2$, respectively, with nonzero coefficients.

*Proof.* Let us express the two single-variable polynomials as

$$f_1(\chi_1) = \sum_p a_p \chi_1^p, \qquad f_2(\chi_2) = \sum_q b_q \chi_2^q.$$

Expanding the product on the left-hand side of Eq. A.2 and applying Lemma A.1 to each term in the product, we get

$$f_1(\chi_1)f_2(\chi_2) = \sum_{p=0}^{\deg(f_1)} \sum_{q=0}^{\deg(f_2)} a_p b_q \sum_{i=1}^{p+q+1} \zeta_{pqi}(\chi_1 + c_{pqi}\chi_2)^{p+q},$$

where subscripts $p$ and $q$ are introduced for $\{c_i, \zeta_i\}$ to elucidate that a different transformation $\boldsymbol{T}$ of dimensions $(p + q + 1) \times 2$ is employed for each monomial. However, due to the freedom of choice in the construction of each $\boldsymbol{T}$, the same $\boldsymbol{T}$ of dimensions $(\deg(f_1) + \deg(f_2) + 1) \times 2$ can be used for all pairwise product terms. This choice stems from the fact that the innermost summation has a final upper limit of $M = \deg(f_1) + \deg(f_2) + 1$. Replacing $p + q + 1$ and $c_{pqi}$ with $M$ and $c_i$, respectively, we can move this summation to the outside:

$$f_1(\chi_1)f_2(\chi_2) = \sum_{i=1}^{M} \sum_{p=0}^{\deg(f_1)} \sum_{q=0}^{\deg(f_2)} a_p b_q \zeta_{pqi}(\chi_1 + c_i\chi_2)^{p+q},$$

resulting in Eq. A.2, where

$$g_i(T_{i1}\chi_1 + T_{i2}\chi_2) = \sum_p \sum_q a_p b_q \zeta_{pqi}(\chi_1 + c_i\chi_2)^{p+q}; \qquad T_{i1} = 1, \qquad T_{i2} = c_i.$$

□

Another consequence is the reproducing property for arbitrary polynomials in two dimensions.

**Lemma A.3.** *For any polynomial $\mathcal{F} = \mathcal{F}(\chi_1, \chi_2)$ of $(\chi_1, \chi_2) \in \mathbb{R}^2$, there exists a linear transformation $\boldsymbol{T}$: $\mathbb{R}^2 \to \mathbb{R}^M$, where $M = \deg(\mathcal{F}) + 1$, and a set of single-variable polynomials $\{g_i: \mathbb{R} \to \mathbb{R}\}_{i=1}^M$ such that*

$$\mathcal{F}(\chi_1, \chi_2) = \sum_{i=1}^{M} g_i(T_{i1}\chi_1 + T_{i2}\chi_2). \tag{A.3}$$

*Proof.* Note that $\deg(\mathcal{F})$ refers to the degree of the multi-variable polynomial, given by the highest degree of the monomials $\{\chi_1^p \chi_2^q\}$ in $\mathcal{F}$. Lemma A.3 can be proved by expanding $\mathcal{F}(\chi_1, \chi_2)$, which produces the same set of monomials as the product of $f_1(\chi_1)$ and $f_2(\chi_2)$ in Lemma A.2, where $\deg(\mathcal{F}) = \deg(f_1) + \deg(f_2)$. This realization concludes the proof. □

Finally, we employ mathematical induction to extend the result to any input dimension.

**Theorem A.1.** *For any polynomial $\mathcal{F} = \mathcal{F}(\boldsymbol{\chi}) = \mathcal{F}(\chi_1, \dots, \chi_N)$ of $\boldsymbol{\chi} \in \mathbb{R}^N$, there exists a linear transformation $\boldsymbol{T}$: $\mathbb{R}^N \to \mathbb{R}^M$, for some $M$ that could be chosen to depend only on $\deg(\mathcal{F})$ and $N$ (e.g., $M = \deg(\mathcal{F}) + N - 1$), and a set of single-variable polynomials $\{g_i: \mathbb{R} \to \mathbb{R}\}_{i=1}^M$ such that*

$$\mathcal{F}(\chi_1, \dots, \chi_N) = \sum_{i=1}^{M} g_i \left( \sum_{j=1}^{N} T_{ij}\chi_j \right). \tag{A.4}$$

*Proof.* Once we have shown that Theorem A.1 holds for a particular input dimension (e.g., $N = 2$), by leveraging the inductive hypothesis to show that it also holds for $N + 1$, the theorem must hold for all subsequent $N$, in accordance with the principle of mathematical induction. For $N = 2$, the polynomial reproducing property has been proved in Lemma A.3.

Assume that Theorem A.1 holds for an arbitrary input dimension $N$. We then consider a polynomial $\mathcal{F} = \mathcal{F}(\chi_1, \dots, \chi_N, \chi_{N+1})$. It can be written as

$$\mathcal{F}(\chi_1, \dots, \chi_N, \chi_{N+1}) = \sum_{p=0}^{P} \mathcal{F}_p(\chi_1, \dots, \chi_N)\chi_{N+1}^p,$$

for some polynomials $\{\mathcal{F}_p\}_{p=0}^P$. By the inductive hypothesis, we have a linear transformation $\boldsymbol{T}_{\mathfrak{N}}$: $\mathbb{R}^N \to \mathbb{R}^{M_{\mathfrak{N}}}$ and a set of polynomials $\{\{\widetilde{g}_{pi}\}_{i=1}^{M_{\mathfrak{N}}}\}_{p=0}^P$ such that

$$\mathcal{F}_p(\chi_1, \dots, \chi_N) = \sum_{i=1}^{M_{\mathfrak{N}}} \widetilde{g}_{pi}(z_{\mathfrak{N},i}), \qquad z_{\mathfrak{N},i} = \sum_{j=1}^{N} T_{\mathfrak{N},ij}\chi_j. \tag{A.5}$$

As a result,

$$\mathcal{F}(\chi_1, \dots, \chi_N, \chi_{N+1}) = \sum_{p=0}^{P} \sum_{i=1}^{M_{\mathfrak{N}}} \widetilde{g}_{pi}(z_{\mathfrak{N},i})\chi_{N+1}^p.$$

By Lemma A.2, for each product term $\widetilde{g}_{pi}(z_{\mathfrak{N},i})\chi_{N+1}^p$ in the summation, there exists a linear transformation $\widetilde{\boldsymbol{T}}_{\mathfrak{N}}$ that maps any pair $(z_{\mathfrak{N},i}, \chi_{N+1})$ to $\mathbb{R}^{M_{\mathfrak{N}+1}}$ such that, for each choice of the indices $p$ and $i$, the product $\widetilde{g}_{pi}(z_{\mathfrak{N},i})\chi_{N+1}^p$ is a sum of single-variable polynomials in the transformed space $\mathbb{R}^{M_{\mathfrak{N}+1}}$, where

$$M_{\mathfrak{N}} + P = M_{\mathfrak{N}+1} = \deg(\mathcal{F}) + N.$$

Composing $\boldsymbol{T}_{\mathfrak{N}}$ together with all $\widetilde{\boldsymbol{T}}_{\mathfrak{N}}$, we get a linear transformation $\boldsymbol{T}_{\mathfrak{N}+1}$ from $\mathbb{R}^{N+1}$ to $\mathbb{R}^{M_{\mathfrak{N}+1}}$ such that $\mathcal{F}(\chi_1, \dots, \chi_N, \chi_{N+1})$ is a linear combination of polynomials $\{g_i\}_{i=1}^{M_{\mathfrak{N}+1}}$ of single variables $\boldsymbol{z} \in \mathbb{R}^{M_{\mathfrak{N}+1}}$. Thus, Theorem A.1 holds for input dimension $N + 1$. This completes the induction process and the proof. □

As noted in Example A.1, due to the nonuniqueness of the transformation, it is possible to achieve more effective and potentially low-dimensional representations through learning and optimization.

## A.2 ADDITIONAL RESULTS FOR KNOT THEORY: CLASSIFICATION

Figure A.1 shows a complete ranking of the inputs and the associated test accuracy of using the top $n$ inputs from one of the runs summarized in Table 2.

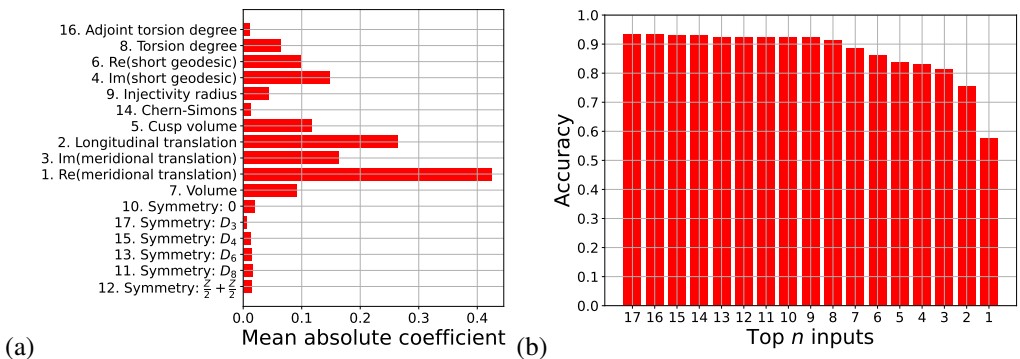

(a)   (b)

Figure A.1: One complete instance of the ranking and accuracy reported in Table 2. (a) Relative importance of the input features based on their mean absolute coefficients, and (b) test accuracy associated with using the top $n$ features in (a).

## A.3 ADDITIONAL RESULTS FOR KNOT THEORY: REGRESSION

Table A.1 compares the performance of the NAM with 17 bases against that of the PNAM with eight bases, both of which can be converted into symbolic equations after training. Considering the poor performance of the NAM, we proceed with just the PNAM. Similar to Table 2, Table A.2 suggests that the meridional translation (real part) and longitudinal translation are the most and second most important invariants, respectively. Nevertheless, solely comparing the mean absolute coefficients may not be sufficient, as unimportant features could be incorrectly ranked as important. We can, however, eliminate these features by examining the accuracy associated with including them, since including an unimportant feature does not improve accuracy compared to the corresponding case with one less input.

Table A.1: Performance of different neural network architectures for the single-output regression problem of predicting the signature of mathematical knots. The mean and standard deviation of the test accuracy are computed from 10 runs.

| Method | Architecture | Parameter count | Test acc. |
|--------|--------------|-----------------|-----------|
| MLP | 3 layers: $[17, 64, 32, 1]$ | $1.2 \times 10^4$ | $94.7 \pm 0.2\%$ |
| NAM | 3 layers: $17 \times [1, 64, 32, 1]$ | $2.1 \times 10^5$ | $65.0 \pm 0.2\%$ |
| PNAM | 3 layers: $8 \times [1, 64, 32, 1]$ | $9.8 \times 10^4$ | $85.5 \pm 0.3\%$ |

Figure A.2 shows a complete ranking of the inputs and the associated test accuracy of using the top $n$ inputs from one of the runs summarized in Table A.2. The ranking in Fig. A.2(d) is determined from the mean absolute coefficients of the linear transformation $\boldsymbol{T}$ in Fig. A.2(b). This linear transformation (i) consists mainly of diagonal elements due to the constraint $\ell_3$ in Eq. 12 and (ii) is sparse due to the constraint $\ell_4$ in Eq. 13. The transformation $\boldsymbol{T}$ with $n = 3$ in Fig. A.2(c) and the scaling coefficients $\boldsymbol{\zeta}$ of just two nonzero bases in Fig. A.2(a) are then leveraged to construct expressions H and I in Table 3. Despite evolving each basis for only one minute using a few basic operators, Fig. A.3 demonstrates that SR can consistently approximate the single-variable bases because of their simple 1D nature. Expressions H and I achieve test accuracy of 81.2 and 81.4%, respectively, comparable to the test accuracy of 81.3% in Fig. A.2(e) for $n = 3$ inputs. Moreover, Table A.3 reveals that both expressions are more than two times faster for inference than a fully connected MLP.

Table A.2: Ranking of important input features based on their mean absolute coefficients for the PNAM in Table A.1. The mean and standard deviation of the test accuracy associated with using only the top $n$ features are computed from their frequency across 10 runs.

| Rank | Input | Symbol | Frequency | Test acc. |
|---|---|---|---|---|
| 1 | Re(meridional translation) | $\chi_{10}$ | 10/10 | $55.7 \pm 1.2\%$ |
| 2 | Longitudinal translation | $\chi_8$ | 6/10 | $74.8 \pm 1.4\%$ |
|  | Torsion degree | $\chi_2$ | 4/10 | $54.9 \pm 0.5\%$ |
| 3 | Torsion degree | $\chi_2$ | 3/10 | $76.4 \pm 0.4\%$ |
|  | Re(short geodesic) | $\chi_3$ | 3/10 | $61.8 \pm 7.3\%$ |
|  | Longitudinal translation | $\chi_8$ | 2/10 | $76.5 \pm 0.2\%$ |
|  | Cusp volume | $\chi_7$ | 1/10 | $81.3\%$ |
|  | Im(short geodesic) | $\chi_4$ | 1/10 | $74.3\%$ |

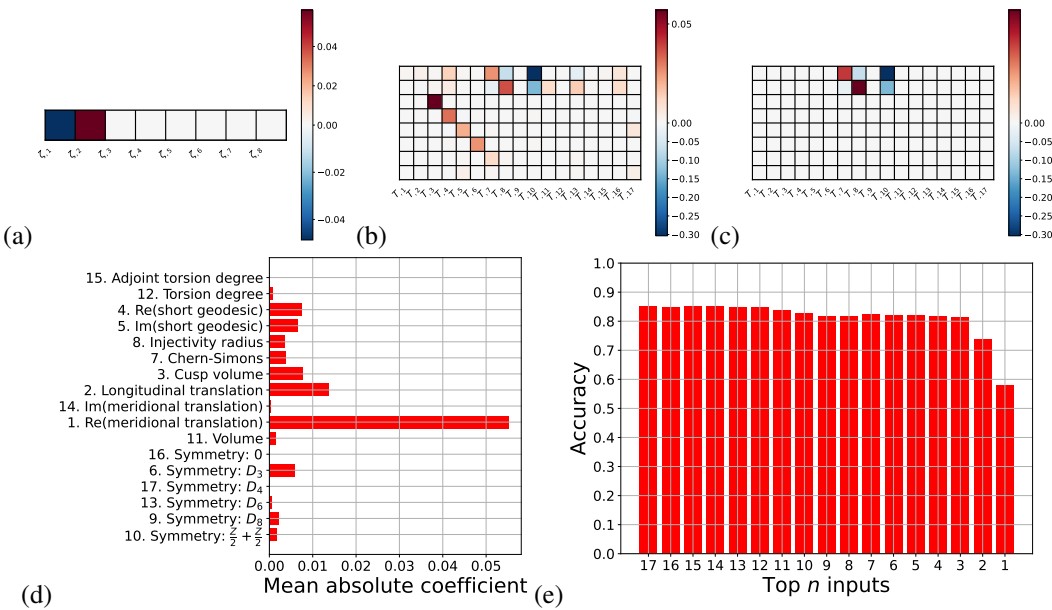

(a)  (b)  (c)  (d)  (e)

Figure A.2: One complete instance of the ranking and accuracy reported in Table A.2, along with optimized parameters of the PNAM. (a) Scaling coefficients $\zeta$ pior to post-processing, (b) linear transformation $T$ pior to post-processing, (c) $T$ with $n = 3$ and rows corresponding to columns of $\zeta$ in (a) having a value of zero set to zero, (d) relative importance of the input features based on their mean absolute coefficients of $T$ in (b), and (e) test accuracy associated with using the top $n$ features in (d).

Recall that the above results correspond to weighting coefficients $w_1 = w_2 = w_3 = w_4 = w_5 = 0.001$, chosen to balance the trade-off between accuracy and sparsity. While increasing the weighting coefficients can make the PNAM sparser, Fig. A.4 indicates that overprioritizing sparsity can lead to less stable training, resulting in poorer performance. Specifically, setting $w_1 = w_2 = w_3 = w_4 = w_5 = 0.01$ results in such a sparse $T$ (see the small coefficient values in Fig. A.5(c)) that the accuracy in Fig. A.5(d) is poor regardless of $n$. On the other hand, Fig. A.5(b) suggests that setting $w_1 = w_2 = w_3 = w_4 = w_5 = 0$, thereby deactivating the regularization constraints, improves (reduces) the accuracy for large (small) $n$ due to the extensive number of co-efficients with large values (the erroneous ranking of the input features) in Fig. A.5(a). Although the exact ranking may not be correct, this model can still narrow down a set of variables (e.g., $n = 6$) that is relevant for predictive accuracy while remaining competitive with the MLP in Table A.1.

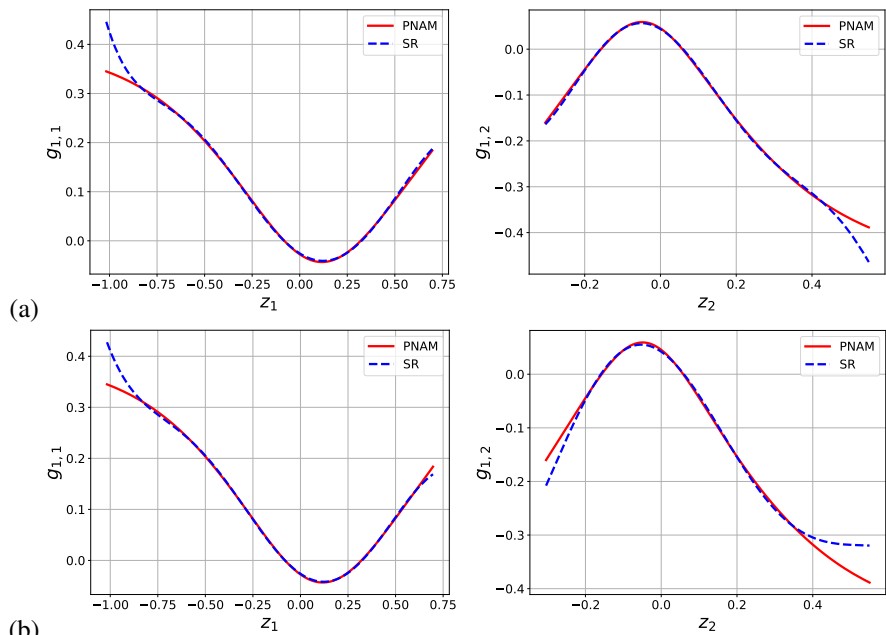

Figure A.3: Reproducibility of SR for learning 1D functions. Both sets of bases (PNAM) are from the run in Fig. A.2, with (a) the two bases (SR) comprising expression H and (b) the two bases (SR) comprising expression I in Table 3. The neural network parameterization of the PNAM, expression H, and expression I all use $n = 3$ features and achieve test accuracy of about 81%.

Table A.3: Training and inference times of the neural networks in Table A.1, along with inference times of expressions G to I in Table 3. Each basis comprising expressions G to I is evolved for one minute. Training times are obtained from an NVIDIA A100-SXM4-40GB GPU, while inference times are obtained from an Apple M1 CPU with 8GB of memory. The test accuracy is repeated to aid comparison.

| Method | Training time (s/epoch) | Inference time (ns/sample) | Test acc. |
|---|---|---|---|
| MLP | 2.1 | 290 | $94.7 \pm 0.2\%$ |
| NAM | 14 | 4300 | $65.0 \pm 0.2\%$ |
| PNAM | 8.9 | 2100 | $85.5 \pm 0.3\%$ |
| Expression G | – | 78 | 75.9% |
| Expression H | – | 130 | 81.2% |
| Expression I | – | 120 | 81.4% |

## A.4 ADDITIONAL RESULTS FOR PHASE FIELD THEORY

In Table A.4, we provide training and inference times of the neural networks in Table 4, in addition to the inference time of the symbolic form in Table 5. As expected, training and inference times of the PNAM increase with the projection dimension $M$. However, similar to Table A.3, converting the PNAM into a compact expression yields faster inference. Since the evaluations of its 1D bases are independent of each other, we plan to parallelize their evaluations in the future to further accelerate training and inference. On the other hand, Fig. A.6 reveals that increasing $M$ improves performance, which may lead to overfitting after saturation. For the most accurate PNAM, we plot and compare its MLP bases with their symbolic approximations (the polynomials $\{g_{ij}\}$ in Table 5), along with their derivatives, in Figs. A.7 and A.8. Test predictions of the average strain energy, phase energy, and axial stress for the linear combinations of the symbolic bases are shown in Fig. A.9.

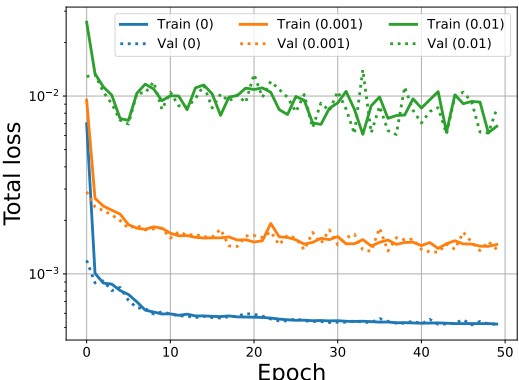

Figure A.4: Effects of weighting coefficients $w_1 = w_2 = w_3 = w_4 = w_5 = \text{a constant}$ on the convergence of the PNAM for the knot data set. Total loss corresponds to the loss function in Eq. 10. The three NAMs use the same projection dimension $M = 8$, and each basis of the PNAMs is an MLP with two hidden layers of 64 and 32 neurons.

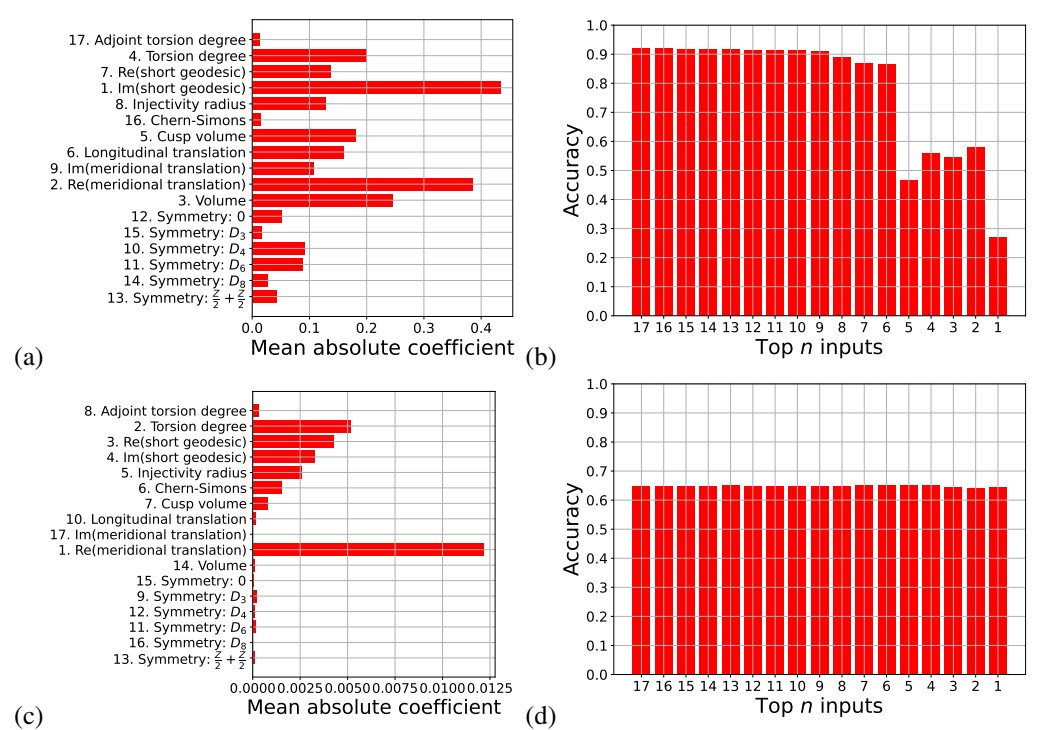

Figure A.5: Effects of weighting coefficients $w_1 = w_2 = w_3 = w_4 = w_5 = \text{a constant}$ on the input ranking and test accuracy of the PNAM for the knot data set. Weighting coefficients $w_1 = w_2 = w_3 = w_4 = w_5 = 0$ for (a) and (b); $w_1 = w_2 = w_3 = w_4 = w_5 = 0.01$ for (c) and (d). (a) and (c): Relative importance of the input features based on their mean absolute coefficients. (b) and (d): Test accuracy associated with using the top $n$ features in (a) and (c), respectively. Results for $w_1 = w_2 = w_3 = w_4 = w_5 = 0.001$ are provided in Fig. A.2(d) and (e). Training and validation histories of the loss function for these three PNAMs are shown in Fig. A.4.

## A.5 HIGH-DIMENSIONAL IMAGE CLASSIFICATION WITH THE MNIST DATA SET

To demonstrate that the PNAM can efficiently handle high-dimensional problems with numerous inputs, we evaluate its performance on the MNIST data set (LeCun et al., 1998), highlighting its di-

Table A.4: Training and inference times of the neural networks in Table 4, along with the inference time of the combined expression (i.e., the sum of the linear combinations of $\{g_{1j}\}$ and $\{g_{2j}\}$) in Table 5. Each nonzero basis $g_{ij}$ is evolved for one minute. Training times are obtained from an NVIDIA A100-SXM4-40GB GPU, while inference times are obtained from an Apple M1 CPU with 8GB of memory. The test MSE is repeated to aid comparison.

| Method | Training time (s/epoch) | Inference time (μs/sample) | Test MSE |
|---|---|---|---|
| MLP | 1.1 | 27 | $(7.05 \pm 2.02) \times 10^{-5}$ |
| NAM | 3.0 | 61 | $(1.12 \pm 0.62) \times 10^{-4}$ |
| PNAM ($M = 3$) | 3.2 | 58 | $(3.31 \pm 2.11) \times 10^{-4}$ |
| PNAM ($M = 8$) | 6.8 | 150 | $(1.34 \pm 0.91) \times 10^{-4}$ |
| PNAM ($M = 16$) | 13 | 320 | $(8.01 \pm 3.76) \times 10^{-5}$ |
| PNAM ($M = 32$) | 26 | 610 | $(6.47 \pm 2.71) \times 10^{-5}$ |
| Expression (Table 5) | – | 17 | $2.25 \times 10^{-5}$ |

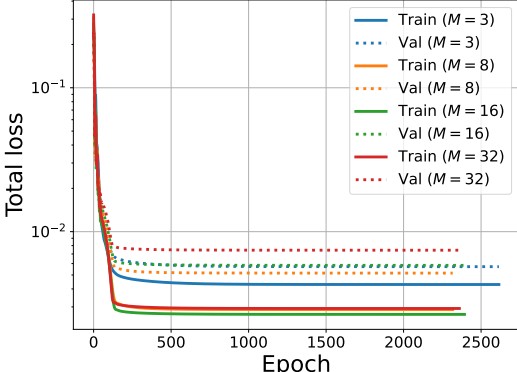

Figure A.6: Effects of the projection dimension $M$ on the convergence of the PNAM for the phase field data set. Total loss corresponds to the loss function in Eq. 10. Each basis of the four PNAMs is an MLP with three hidden layers of 256 neurons; all models are trained using weighting coefficients $w_1 = w_2 = w_3 = w_4 = w_5 = 0.001$.

mensionality reduction capability and providing visual insight into its input pruning mechanism. The MNIST data set contains 70,000 images of handwritten digits from 0 to 9, each of size $28 \times 28$ pixels, partitioned into 60,000 images for training/validation and 10,000 images for testing. Each image is reshaped into a 784-dimensional feature vector and normalized, resulting in a high-dimensional classification task where the model must map dense feature vectors to their corresponding digit labels. This experiment illustrates the ability of the PNAM to preserve interpretability while achieving strong predictive accuracy by identifying and utilizing only the most informative inputs, effectively pruning irrelevant features.

In Table A.5, we compare PNAMs against a standard NAM and conventional MLPs, all of which are trained for up to 1000 epochs with an early stopping patience of 50 epochs. First, the linear transformation $\boldsymbol{T}$ allows us to project the original 784-dimensional vector into a lower-dimensional feature space of size $M$. Even with a small $M = 8$, the resulting PNAM outperforms the standard NAM, despite the latter requiring one basis per input and having nearly 100 times more parameters. Second, increasing $M$ enhances the expressivity of the PNAM, allowing it to fit the training data perfectly (see Fig. A.10) and achieve a test accuracy of 98.1% for $M = 64$. Third, although PNAMs perform similarly to MLPs, suggesting comparable expressive power in practice, they retain the key advantage of interpretability. Unlike black-box MLPs, PNAMs offer insight into which variables are crucial for predictive accuracy, enabling model transparency and post hoc analysis.

Figure A.11 illustrates that the PNAM produces a consistent ranking of the feature importance across different values of $M$, demonstrating robustness in its learned feature sensitivity. This consistency

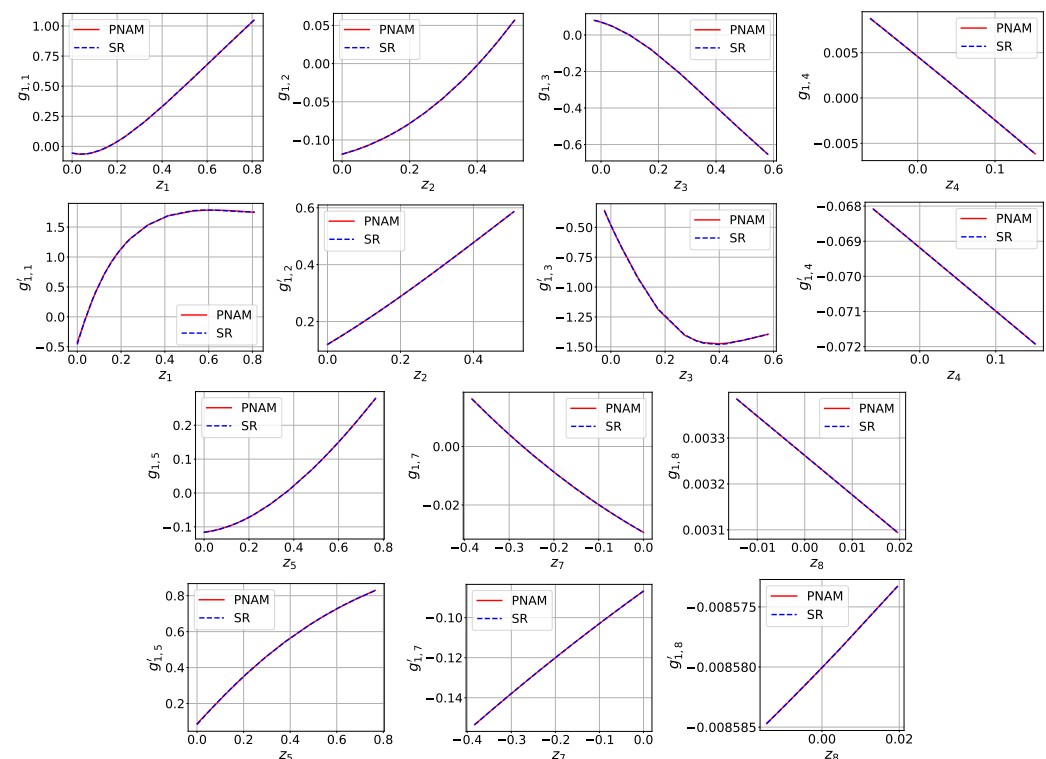

Figure A.7: Bases $\{g_{1j}\}$ and their derivatives. The expressions of $\{g_{1j}\}$ are presented in Table 5.

Table A.5: Performance of different neural network architectures for the multi-label classification problem of predicting handwritten digits from the MNIST data set. Training and validation histories of the accuracy for the PNAMs are provided in Fig. A.10.

| Method | Architecture | Parameter count | Training acc. | Test acc. |
|---|---|---|---|---|
| MLP | 3 layers: [784, 64, 32, 10] | $1.2 \times 10^4$ | 99.9% | 96.1% |
| MLP | 3 layers: [784, 128, 64, 10] | $4.9 \times 10^4$ | 99.9% | 96.3% |
| MLP | 4 layers: [784, 128, 64, 32, 10] | $6.6 \times 10^4$ | 99.9% | 96.7% |
| MLP | 4 layers: [784, 256, 128, 64, 10] | $2.6 \times 10^5$ | 99.9% | 97.1% |
| NAM | 3 layers: $784 \times [1, 64, 32, 10]$ | $9.6 \times 10^6$ | 95.0% | 93.0% |
| PNAM | 3 layers: $8 \times [1, 64, 32, 10]$ | $1.0 \times 10^5$ | 96.0% | 94.7% |
| PNAM | 3 layers: $16 \times [1, 64, 32, 10]$ | $2.1 \times 10^5$ | 98.5% | 96.7% |
| PNAM | 3 layers: $32 \times [1, 64, 32, 10]$ | $4.2 \times 10^5$ | 100% | 97.5% |
| PNAM | 3 layers: $64 \times [1, 64, 32, 10]$ | $8.4 \times 10^5$ | 100% | 98.1% |

supports effective pruning of nonessential inputs with minimal impact on accuracy. Specifically, for the PNAM with $M = 64$, we remove the least important pixels and visualize two progressively smaller subsets in Fig. A.12. Keeping only the top 400 and 200 relevant pixels, this model still achieves test accuracy of 98.0 and 94.0%, respectively. Visual inspection of the remaining pixels reveals that they occupy semantically meaningful regions, allowing human observers to easily identify the underlying digit classes. Further insight is provided by Fig. A.13, which shows that the majority of predictive information is concentrated in a small number of dominant singular values, enabling both dimensionality reduction and feature pruning.

Finally, we examine the impact of weighting coefficients $w_1, w_2, \ldots, w_5$ on the performance of the PNAM. Similar to the trend observed in Fig. A.4, Fig. A.14 indicates that increasing the weighting coefficients leads to less stable training as the optimization objectives shift from prioritizing predictive accuracy to enforcing sparsity. Nevertheless, as shown in Table A.6, these weighting co-

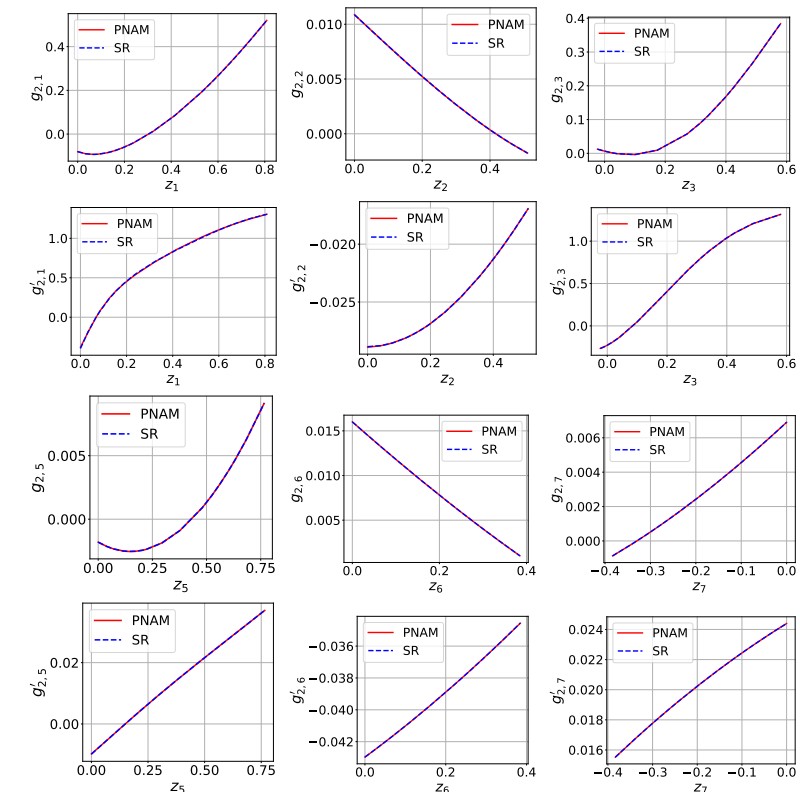

Figure A.8: Bases $\{g_{2j}\}$ and their derivatives. The expressions of $\{g_{2j}\}$ are presented in Table 5.

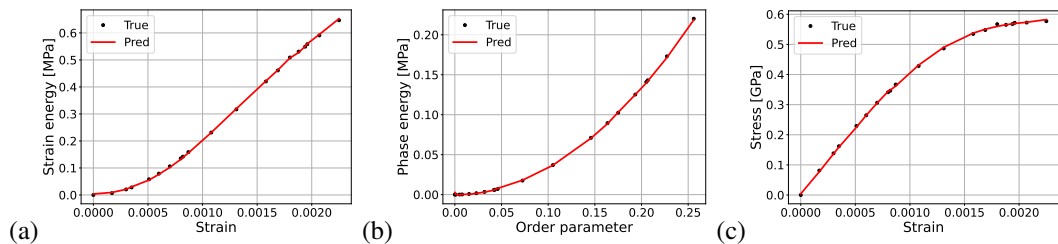

Figure A.9: Test predictions for the linear combinations of the polynomials $\{g_{ij}\}$ in Table 5. (a) Linear combination of $\{g_{1j}\}$, (b) linear combination of $\{g_{2j}\}$, and (c) derivative of the sum of the linear combinations of $\{g_{1j}\}$ and $\{g_{2j}\}$ with respect to the strain.

efficients can be tuned to obtain a desired trade-off between accuracy and sparsity, tailored to the specific needs of the users.

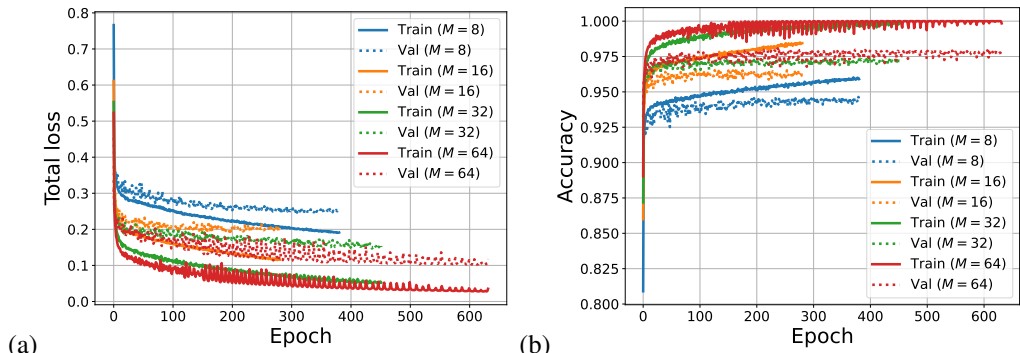

(a)          (b)

Figure A.10: Effects of the projection dimension $M$ on the convergence and accuracy of the PNAM for the MNIST data set. Training and validation histories of (a) the loss function in Eq. 10 and (b) accuracy. Each basis of the four PNAMs is an MLP with two hidden layers of 64 and 32 neurons; all models are trained using weighting coefficients $w_1 = w_2 = w_3 = w_4 = w_5 = 0.01$.

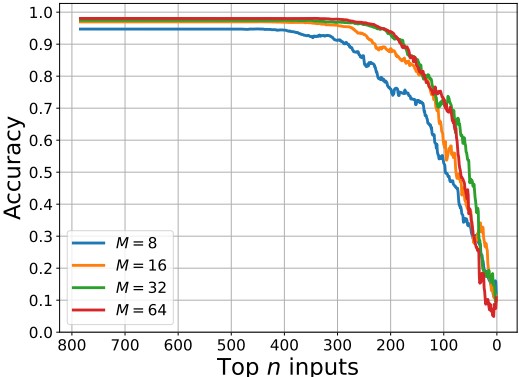

Figure A.11: Impact of input pruning on the test accuracy for the PNAMs in Table A.5. Unimportant inputs are pruned according to their mean absolute coefficients (see Section 2.5).

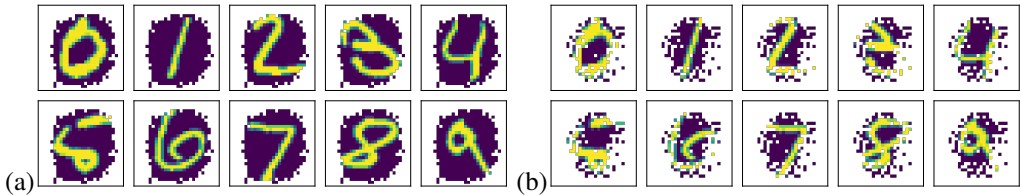

(a)          (b)

Figure A.12: Demonstration of important inputs for the PNAM with projection dimension $M = 64$ in Fig. A.11. Shown are the first 10 unique handwritten digits in the MNIST test set, arranged from 0 to 9, using (a) $n = 400$ ($\sim$50%) and (b) $n = 200$ ($\sim$25%) most relevant pixels, which achieve test accuracy of 98.0 and 94.0%, respectively.

Table A.6: Trade-off between accuracy and sparsity due to weighting coefficients $w_1 = w_2 = w_3 = w_4 = w_5 =$ a constant for the PNAMs in Fig. A.14. An element $T_{jk} < 10^{-4}$ is considered zero, since setting such an element to zero has little to no effect on the test accuracy.

| Weighting coefficients | Training acc. | Test acc. | Percent of zero $T_{jk}$ |
|---|---|---|---|
| 0.001 | 95.6% | 94.3% | 14.1% |
| 0.01 | 96.0% | 94.7% | 32.8% |
| 0.1 | 95.7% | 94.7% | 68.9% |
| 1 | 90.9% | 91.3% | 76.5% |

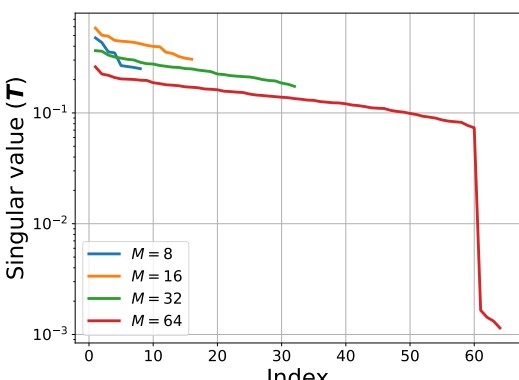

Figure A.13: Singular values of the linear transformation $T$ for the PNAMs in Table A.5. The linear decay and sudden drop in magnitude of the singular values on a log scale suggest that we can prune coefficients in $T$ without severely degrading performance.

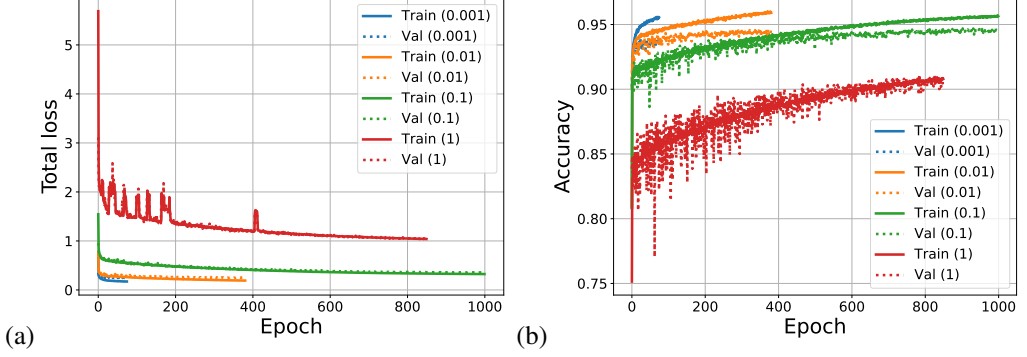

(a)                      (b)

Figure A.14: Effects of weighting coefficients $w_1 = w_2 = w_3 = w_4 = w_5 = \text{a constant}$ on the convergence and accuracy of the PNAM for the MNIST data set. Training and validation histories of (a) the loss function in Eq. 10 and (b) accuracy. The four NAMs use the same projection dimension $M = 8$, and each basis of the PNAMs is an MLP with two hidden layers of 64 and 32 neurons.

