# OpenReview forum: "Projected Neural Additive Models as Universal Approximators"
_ICLR.cc/2026/Conference — Submitted to ICLR 2026_

### Official Review · Reviewer_9BQQ · 2025-10-17

**Soundness:** 2
**Presentation:** 3
**Contribution:** 2
**Rating:** 4
**Confidence:** 4

**Summary:**

This paper introduces Projected Neural Additive Models (PNAM), an extension of Neural Additive Models (NAM) that achieves universal approximation capability by incorporating a linear transformation of inputs before processing them through independent single-variable neural networks. The authors establish the theoretical foundation for PNAM's universal approximation property using the Stone-Weierstrass theorem and propose regularization and post-processing techniques to enhance interpretability. Through experiments on mathematical knot invariants and phase field simulations, they demonstrate PNAM's competitive performance against MLPs and NAMs while maintaining better interpretability. The work is significant as it provides a balance between expressivity and interpretability, particularly beneficial for scientific domains where understanding model behavior is crucial.

**Strengths:**

The paper presents several notable strengths. First, it provides a rigorous theoretical foundation by formally proving PNAM's universal approximation property using the Stone-Weierstrass theorem, addressing a key limitation of the original NAM. Second, the proposed architecture elegantly combines the interpretability of additive models with enhanced expressivity through input projection, representing a meaningful advancement in interpretable neural network design. Third, the comprehensive regularization framework—including weight decay, function value constraints, input coupling penalties, and sparsity promotion—effectively addresses the trade-off between model complexity and interpretability. Fourth, the post-processing techniques for feature importance ranking, parameter pruning, and symbolic regression conversion offer valuable tools for enhancing model transparency. Finally, the experimental evaluation on two distinct domains (knot theory and phase field fracture) demonstrates the model's versatility and provides convincing evidence of its competitive performance against relevant baselines.

**Weaknesses:**

1) While the paper presents a compelling approach, there are several aspects that warrant further consideration. First, the theoretical foundation primarily focuses on the universal approximation property, but lacks analysis of the optimization process. Specifically, there is no discussion of convergence guarantees for the Adam optimizer when applied to PNAM's architecture, nor is there an analysis of the convexity properties of the loss function with the proposed regularization terms. Additionally, the impact of the projection dimension M on the optimization landscape and convergence speed remains unexplored.

2) The experimental setup could benefit from more comprehensive details. The selection of regularization weights (w1-w5) is not adequately justified, as they are simply set to fixed values without sensitivity analysis. Similarly, the criteria for choosing key hyperparameters such as the projection dimension M and network architecture are not well explained. Moreover, the paper lacks information on computational requirements, training times, and resource consumption, which are important for assessing practical feasibility.

3) The scale and diversity of the experimental data raise some concerns. While the knot theory dataset is substantial, it only has 17 input dimensions, which may not represent high-dimensional challenges. The phase field dataset, on the other hand, is extremely small (only 96 samples), which may lead to overfitting and limit generalization. Furthermore, the absence of experiments on truly high-dimensional, large-scale datasets or standard machine learning benchmarks makes it difficult to evaluate PNAM's performance in more realistic settings.

4) The comparison with alternative approaches is somewhat limited. The paper primarily benchmarks against MLP, NAM, and KAN, but omits comparisons with other interpretable neural network methods and traditional statistical approaches like Generalized Additive Models (GAMs). Additionally, the comparison with symbolic regression methods is insufficient, especially given the emphasis on converting PNAM to symbolic form.

5) There is a lack of systematic parameter sensitivity analysis. The paper does not explore how the projection dimension M affects model performance and interpretability in depth, nor does it analyze the sensitivity to regularization weights, despite their critical role in balancing accuracy and interpretability. The impact of network architecture choices (depth, width) on performance is also not adequately addressed. Moreover, the evaluation of the quality and reliability of the symbolic expressions obtained through post-processing is limited, and there is no analysis of PNAM's computational complexity during training and inference.

**Questions:**

Please refer to the above questions.

It's a fascinating insight for NAMs, while the proof seems largely similar to generalized additive models.
I would be grateful and willing to raise my scores if the authors would solve my above concerns.

---

> ### Author Response · Authors · 2025-11-24
> **Round 1, Part 1**
>
> We thank the reviewer for their thorough review and constructive suggestions. Before responding to Weaknesses, we would like to respond to Questions to clarify any misunderstandings that may have contributed to the weaknesses.
>
> $\textbf{Questions:}$
>
> It's a fascinating insight for NAMs, while the proof seems largely similar to generalized additive models. I would be grateful and willing to raise my scores if the authors would solve my above concerns.
>
> $\textbf{Response to Questions:}$
>
> Can the reviewer point to the paper(s) in which the proof of universal approximation for generalized additive models (GAMs) is provided? To the best of our knowledge, GAMs, which NAMs are (see Eq. 1 of https://arxiv.org/pdf/2004.13912), are linear combinations of nonlinear functions of individual inputs. By counterexample, one can show that standard GAMs cannot reproduce functions with coupling terms, such as $\mathcal{F}(\chi_1, \chi_2) = \chi_1 \chi_2$. While $\chi_1 \chi_2$ can be approximated with a GA$^2$M (e.g., see Eq. 1 of https://arxiv.org/pdf/2205.14120), it is easy to see how the number of feature functions of GAMs would quickly blow up if one needs to introduce increasingly higher-order coupling functions to handle problems with numerous inputs and complex interactions. On the other hand, PNAMs are linear combinations of nonlinear functions of projected inputs. As an illustrative example, we show in Example A.1 of our paper that PNAMs can elegantly reproduce $\chi_1 \chi_2$. We further prove, with mathematical induction, that PNAMs can reproduce all polynomials, which are dense in the space of continuous functions by the Stone--Weierstrass theorem. Thus, PNAMs are universal approximators and possess the expressive power to approximate any continuous function, provided that the optimization process is successful.
>
> We are currently working on a third numerical experiment using the MNIST data set (to be updated in the Appendix) as a high-dimensional problem, while also generating results to address the concerns in Weaknesses. We will respond again once these results are ready.

---

> > ### Author Response · Authors · 2025-11-27
> > **Round 1, Part 2**
> >
> > $\textbf{Weaknesses:}$
> >
> > $\textit{\textbf{W1.}}$ While the paper presents a compelling approach, there are several aspects that warrant further consideration. First, the theoretical foundation primarily focuses on the universal approximation property, but lacks analysis of the optimization process. Specifically, there is no discussion of convergence guarantees for the Adam optimizer when applied to PNAM's architecture, nor is there an analysis of the convexity properties of the loss function with the proposed regularization terms. Additionally, the impact of the projection dimension M on the optimization landscape and convergence speed remains unexplored.
> >
> > $\textbf{Response to W1.}$ Convergence guarantees for Adam and other gradient-based optimizers have been presented in https://arxiv.org/pdf/2202.12482, specifically Section 3.4, which proves convergence for the (sparse) NAM using group sparsity, i.e., regularization constraint $\ell_5$ in our paper. This analysis applies to the PNAM, and a similar strategy can be used for the other four constraints. Nevertheless, we admit introducing additional constraints makes the optimization landscape more difficult to traverse (see, e.g., https://www.sciencedirect.com/science/article/pii/S002199912100663X). Therefore, in future studies, we plan to leverage techniques, such as those introduced in https://arxiv.org/pdf/1712.09913, to visualize the loss landscape and make it smoother/better-behaved. In the meantime, we present in Figs. A.4, A.6, A.10, and A.14 effects of the weighting coefficients in Eq. 10 and/or the projection dimension $M$ on the convergence of the PNAM for three numerical experiments. The PNAM converges to (local or global) minima regardless of the weighting coefficients and $M$.
> >
> > Figures A.2, A.4, and A.5 (knot experiment) and Fig. A.14 and Table A.6 (MNIST experiment) show the trade-off between accuracy and sparsity due to the weighting coefficients. We emphasize that the regularization constraints are additional tools that we introduce for users to make the neural network model sparser and the symbolic expression shorter. If one does not care about this aspect, they can simply set the weighting coefficients to zero, which actually increases performance and improves convergence. Figure A.6 (phase field experiment) and Fig. A.10 (MNIST experiment) demonstrate that increasing $M$ provides the PNAM with higher expressivity, which saturates in the phase field experiment and enables the model to perfectly fit the training data in the MNIST experiment. Similar to conventional deep neural networks, this higher expressivity can increase performance for unseen data or may lead to overfitting. For additional studies on the effects of $M$, we point the reviewer to Figs. 5, 7, 9, and 14 in https://www.sciencedirect.com/science/article/pii/S0045782525000647.

---

> > > ### Author Response · Authors · 2025-11-27
> > > **Round 1, Part 3**
> > >
> > > $\textit{\textbf{W2.}}$ The experimental setup could benefit from more comprehensive details. The selection of regularization weights (w1-w5) is not adequately justified, as they are simply set to fixed values without sensitivity analysis. Similarly, the criteria for choosing key hyperparameters such as the projection dimension M and network architecture are not well explained. Moreover, the paper lacks information on computational requirements, training times, and resource consumption, which are important for assessing practical feasibility.
> > >
> > > $\textbf{Response to W2.}$ We have provided additional results for the weighting coefficients, projection dimension $M$, training times, and inference times. For the weighting coefficients and $M$, see our response to W1. Additionally, we note that Eq. 10 and regularization constraints $\ell_1, \ell_2, \dots, \ell_5$ are formulated such that the (expanded) loss terms in Eq. 10 are similar in magnitude (e.g., $\ell_1$ using $|\cdot||_2$ as opposed to $||\cdot||_2^2$, $\ell_1$ and $\ell_2$ scaled by 1 over the number of data points $D$, and $\ell_3$, $\ell_4$, and $\ell_5$ scaled by 1 over $M$). While our framework and code are set up to allow for individual selections of $w_1, w_2, \dots, w_5$, this formulation enables them to be selected together, thereby reducing the computational cost associated with hyperparameter tuning. Furthermore, we found that setting $w_1 = w_2 = w_3 = w_4 = w_5 = 0.01$ and $w_1 = w_2 = w_3 = w_4 = w_5 = 0.001$ for classification and regression problems, respectively, provides a good initial trade-off between accuracy and sparsity due the values of $\mathcal{L}$ in Eq. 10 for scaled variables in relation to the other loss terms. The weighting coefficients can be increased or even set to zero, depending on the needs of the users.
> > >
> > > Regarding computational requirements, training times, and resource consumption, we have updated footnote 6 to correctly estimate the number of parameters of PNAMs for high-dimensional problems. If the number of inputs $N$ is significantly larger than $W$, the number of neurons in the widest layer, then the projection matrix will contribute meaningfully to the parameter count. (This update does not change the parameter count for the knot and phase field problems.) We have also added a sentence in footnote 6 to mention the relationship between the number of parameters and memory requirements. Compared to an MLP that uses the same number of layers $L$ and $W$, the PNAM has approximately $M$ times more parameters. Considering that memory requirements scale linearly with the number of parameters, the PNAM requires $M$ times more memory and is thus slower to train than the MLP (see Tables A.3 and A.4). However, the PNAM, after conversion to symbolic expressions, can be faster for inference than the MLP, agreeing with the findings in https://www.sciencedirect.com/science/article/pii/S0045782525000647.
> > >
> > > Due to the relationship between $M$ and the training time, we recommend starting with $M = 8$ or a square projection (whichever is smaller) and adjusting $M$ as needed. As for network architecture, the width and depth of the 1D MLPs are analogous to those of conventional MLPs and are problem-specific. In the future, we plan to parallelize the evaluations of the 1D bases (since they are independent of each other), which should reduce training times and further accelerate inference.

---

> > > > ### Author Response · Authors · 2025-11-27
> > > > **Round 1, Part 4**
> > > >
> > > > $\textit{\textbf{W3.}}$ The scale and diversity of the experimental data raise some concerns. While the knot theory dataset is substantial, it only has 17 input dimensions, which may not represent high-dimensional challenges. The phase field dataset, on the other hand, is extremely small (only 96 samples), which may lead to overfitting and limit generalization. Furthermore, the absence of experiments on truly high-dimensional, large-scale datasets or standard machine learning benchmarks makes it difficult to evaluate PNAM's performance in more realistic settings.
> > > >
> > > > $\textbf{Response to W3.}$ Our goal with the PNAM is to apply it to physical sciences, which is why we experiment with invariants in knot theory and phase field fracture mechanics. As the reviewer points out, the knot data set is substantial; on the other hand, we purposely use a small phase field data set to reflect real-world engineering challenges where data are often limited and lacking. We, however, thank the reviewer for bringing up the concern regarding high-dimensional problems; it has led us to conduct one more numerical experiment using the MNIST data set. We present our results for this standard machine learning benchmark in Appendix A.5 (with details of the experiment to be filled in). For background, the MNIST data set contains 70,000 images (60,000 for training/validation and 10,000 for testing) of $28 \times 28$ pixels. By reshaping the images into vectors of 784 pixels/features, we use this high-dimensional classification problem to illustrate the advantages and mechanisms of PNAMs and compare with vanilla NAMs and fully connected MLPs, all set to train for 1000 epochs using an early stopping patience of 50 epochs.
> > > >
> > > > First and foremost, the linear transformation allows us to project the 784 features onto a smaller feature space. Simply using $M = 8$, the PNAM outperforms the NAM, which requires one basis for every feature and has approximately 100 times more parameters (see Table A.5). Second, increasing $M$ provides the PNAM with higher expressivity, enabling it to perfectly fit the training data and attain a test accuracy of 98.1\% for $M = 64$. Third, while PNAMs perform similarly to MLPs (i.e., additive models can now achieve the same expressivity as fully connected neural networks), PNAMs provide insight into which features are important for predictive accuracy that MLPs cannot. In Fig. A.12, we visualize the input pruning mechanism of the PNAM. Using only $\sim$25\% of the pixels, the PNAM with $M = 64$ still achieves a test accuracy of 94.0\%. Furthermore, we as humans are able to verify that the inputs the PNAM deems important are indeed important, as we can still discern which handwritten digit the remaining pixels represent.

---

> > > > > ### Author Response · Authors · 2025-11-27
> > > > > **Round 1, Part 5**
> > > > >
> > > > > $\textit{\textbf{W4.}}$ The comparison with alternative approaches is somewhat limited. The paper primarily benchmarks against MLP, NAM, and KAN, but omits comparisons with other interpretable neural network methods and traditional statistical approaches like Generalized Additive Models (GAMs). Additionally, the comparison with symbolic regression methods is insufficient, especially given the emphasis on converting PNAM to symbolic form.
> > > > >
> > > > > $\textbf{Response to W4.}$ We apologize for not making the distinctions between the PNAM and GAMs or classical SR algorithms clearer. We have attempted to make their distinctions clearer by expanding the Introduction and adding a Remark in Section 2.1. First, we compared PNAMs with MLPs and KANs because of their established universal approximation capabilities. In contrast, standard GAMs are not universal approximators. For comparison of the NAM, which is a GAM and a model that the PNAM reverts to if the linear transformation is a square identity matrix, with other interpretable methods and GAMs, see Table 1 of https://arxiv.org/pdf/2004.13912, which shows that the NAM performs similarly to other interpretable methods and GAMs for simple problems. Nevertheless, they all underperform MLPs for more complex problems, such as the one using the California housing data set. By including NAMs in our comparison with MLPs and KANs, we demonstrate that PNAMs can close the performance gap exhibited by NAMs (and other GAMs).
> > > > >
> > > > > Regarding comparison with classical SR algorithms, we emphasize here (and also in Related work) that although we have equipped the PNAM with the ability to produce mathematical expressions, SR is primarily employed for pruning to reduce the number of parameters and to gain insight into the interactions of the input variables. The additive nature of PNAMs and other additive models, such as KANs, prevents them from recovering compact equations and concisely reproducing operators like division (see, e.g., Fig. 3 of https://openreview.net/pdf?id=TqzNI4v9DT). On the other hand, classical SR algorithms, such as AI Feynman, are designed for recovering compact physical laws and perform poorly on problems without precise functional forms (e.g., compare Figs. 1 and 3 in https://arxiv.org/pdf/2107.14351). Considering that PNAMs and classical SR algorithms are designed for different purposes, we did not try to compare them.
> > > > >
> > > > > $\textit{\textbf{W5.}}$ There is a lack of systematic parameter sensitivity analysis. The paper does not explore how the projection dimension M affects model performance and interpretability in depth, nor does it analyze the sensitivity to regularization weights, despite their critical role in balancing accuracy and interpretability. The impact of network architecture choices (depth, width) on performance is also not adequately addressed. Moreover, the evaluation of the quality and reliability of the symbolic expressions obtained through post-processing is limited, and there is no analysis of PNAM's computational complexity during training and inference.
> > > > >
> > > > > $\textbf{Response to W5.}$ For discussions on hyperparameter sensitivity and computational complexity, see our responses to W1 and W2. Additionally, in Fig. A.11, we demonstrate that the projection dimension $M$ has a minimal impact on the ranking of the input features for predictive accuracy. Moreover, while the width and depth of the 1D MLPs are problem-specific, we recommend starting with small MLPs, such as those using two hidden layers of 64 and 32 neurons, and increasing the width and depth of the MLPs as needed. Regarding the quality and reliability of the symbolic expressions, we refer the reviewer to Figs. A.7 and A.8 for the phase field experiment, which learns expressions that approximate the 1D MLPs almost perfectly. In addition, we have added Fig. A.3 to illustrate the different symbolic bases that comprise Formulas H and I in Table 3, both of which achieve similar accuracy to their neural network parameterization for $n = 3$ variables. Recall that each basis is only evolved for one minute. With a longer evolution time and more operators to choose from, SR can better approximate the 1D MLPs, although doing so for the knot experiment would not further improve the test accuracy.

---

### Official Review · Reviewer_aJnm · 2025-10-31

**Soundness:** 2
**Presentation:** 3
**Contribution:** 2
**Rating:** 2
**Confidence:** 3

**Summary:**

This study introduces a method to enhance the expressivity of neural additive models (NAMs) while allowing for interpretability. Specifically, it proves that the projected neural additive model (PNAM), which extends NAMs by applying a learnable linear transformation to the features before they enter the additive structure, achieves the universal approximation property. To address the reduced interpretability that arises from feature coupling after the linear transformation, the autors introduce regularization strategies that promote sparsity and penalize unnecessary interactions, enabling ranking and pruning of features and transformations. Finally, symbolic regression converts the pruned MLPs into mathematical expressions, further enhancing interpretability.

**Strengths:**

● Theoretical support for universal approximation. The paper makes a theoretical contribution by establishing the universal approximation property of PNAM, which had not been previously proven.

● Efforts to enhance interpretability. The framework employs both regularization and post-hoc techniques to improve interpretability by encouraging the use of only the necessary linear transformations of features. Moreover, converting the pruned MLPs into mathematical expressions enhances interpretability while reducing computational complexity.

**Weaknesses:**

● Reduced interpretability due to feature coupling. The learnable linear transformation entangles multiple features. This makes it difficult to attribute model behavior to individual inputs and limits interpretability.

● Computational complexity. The additional optimization required for regularization tuning and symbolic regression may introduce significant computational overhead.

● Dependence on hyperparameter choices. The performance and sparsity outcomes appear sensitive to the selection of projection dimension and regularization weights, requiring extensive tuning.

● Lack of synthetic experiments for interpretability validation. Synthetic experiments with known ground truth would enable a clearer quantitative assessment of interpretability, but such evaluation is currently missing.

● Limited evaluation of pruning and symbolic regression. Given the performance gap between the symbolic expression and the original MLP, it would strengthen the contribution to clearly justify why the derived symbolic form can still be regarded as interpretable and reliable.

**Questions:**

● Could you clarify the column descriptions in Table 3? In particular, it would be helpful to explain the distinction between the “reported” and “ours” values in the evaluation of test accuracy, the reason for the large discrepancy between them, and the precise definition of “Total acc.”

---

> ### Author Response · Authors · 2025-11-24
> **Round 1, Part 1**
>
> We thank the reviewer for their critical evaluation and helpful insights. We will respond to Questions first.
>
> $\textbf{Questions:}$
>
> $\textbf{Q1.}$ Could you clarify the column descriptions in Table 3? In particular, it would be helpful to explain the distinction between the "reported" and "our" values in the evaluation of test accuracy, the reason for the large discrepancy between them, and the precise definition of "Total acc."
>
> $\textbf{Response to Q1.}$ Yes, we have updated the following sentence in Section 3.1 (Regression).
>
> "Since the accuracy appears to depend on the test set that results from a random data split, we evaluate all formulas on our test set and the entire knot data set (denoted as "Our" and "Total acc." in Table 3), in addition to the values reported by Liu et al. (2024)."
>
> Please also see Supplementary Material, which includes a Jupyter Notebook that reproduces the results for "Total acc." Regarding the large discrepancy between them, we will refrain from making any speculation. We are simply reporting our evaluated results based on the formulas provided in Table 4 of Liu et al. (2024).

---

> > ### Author Response · Authors · 2025-11-28
> > **Round 1, Part 2**
> >
> > $\textbf{Weaknesses:}$
> >
> > $\textbf{W1.}$ Reduced interpretability due to feature coupling. The learnable linear transformation entangles multiple features. This makes it difficult to attribute model behavior to individual inputs and limits interpretability.
> >
> > $\textbf{Response to W1.}$ We acknowledge that the PNAM alters the direct feature-to-output mapping found in standard NAMs. However, we view this not as a loss of interpretability, but rather as a necessary trade-off to achieve the expressivity required for capturing many physical and scientific problems. Standard NAMs prioritize feature attribution (i.e., isolating the contributions of $\\{\chi_i\\}$), but this strict additivity often fails to capture fundamental interactions ubiquitous in scientific laws. Consider Newton's second law $F = m a$. The physical mechanism is inherently multiplicative, meaning the effect of the force $F$ on the acceleration $a$ depends entirely on the mass $m$. A standard NAM is mathematically restricted to an approximation of the form $g_1(m) + g_2(a)$, which essentially assumes the variables act independently---an assumption that is interpretable but physically incorrect for this system.
> >
> > The PNAM addresses this limitation by shifting the interpretability goal from isolating variables to discovering structure, first by revealing which variables are crucial for predictive accuracy (see, e.g., Figs. A.1 and A.12). Furthermore, by employing the projection layer, the PNAM captures necessary couplings (e.g., mixing $m$ and $a$ to find their interaction), while our sparsity regularization constraints ensure that only physically relevant couplings are retained. When combined with the (optional) symbolic recovery step described in Section 2.5, these tools allow the model to output explicit mathematical relationships (e.g., recovering interaction terms), rather than just a feature importance score, thereby providing a deeper level of scientific interpretability.
> >
> > $\textbf{W2.}$ Computational complexity. The additional optimization required for regularization tuning and symbolic regression may introduce significant computational overhead.
> >
> > $\textbf{Response to W2.}$ We acknowledge that the processes of regularization tuning and symbolic regression (SR) could add computational cost. However, we emphasize that these components are optional tools designed to give users flexibility in balancing model accuracy, sparsity, and interpretability according to their specific needs. Crucially, the regularization terms can be deactivated (i.e., weights set to zero) when sparsity is not a priority, effectively reducing the model to a standard, accuracy-optimized configuration. To mitigate the cost of regularization tuning, we have carefully formulated the loss function (Eq. 10) so that the regularization weights can be selected together. This design significantly simplifies the search space. Furthermore, based on the relative scale of $\mathcal{L}$ in Eq. 10 for normalized variables in relation to the regularization constraints, we provide empirically validated default values for the regularization weights that yield a robust initial trade-off between accuracy and sparsity.
> >
> > Regarding SR, it is computationally lightweight in our setup. Each 1D basis is evolved for only one minute, which is feasible due to the simplicity and low dimensionality of the individual regressions. Despite their short evolution time, the resulting symbolic expressions accurately approximate their neural network parameterizations (see Figs. A.3, A.7, and A.8). Moreover, replacing the learned MLP bases with closed-form symbolic expressions accelerates inference, as demonstrated in Tables A.3 and A.4. In summary, while some overhead exists, we have implemented several design choices to minimize it. We emphasize that all sparsification steps are optional, allowing users to adapt the pipeline to their computational needs.

---

> > > ### Author Response · Authors · 2025-11-28
> > > **Round 1, Part 3**
> > >
> > > $\textbf{W3.}$ Dependence on hyperparameter choices. The performance and sparsity outcomes appear sensitive to the selection of projection dimension and regularization weights, requiring extensive tuning.
> > >
> > > $\textbf{Response to W3.}$ We agree that, like any flexible neural architecture (including standard MLPs and KANs), the PNAM requires hyperparameter selection, specifically for the projection dimension $M$ and regularization weights. Nevertheless, we view this not as a limitation, but as an explicit feature that allows users to control the complexity-interpretability trade-off. The projection dimension $M$ defines the maximum rank of interactions the model can capture. In practice, we find that this sensitivity is manageable and follows intuitive patterns (e.g., performance improves with $M$ until saturation). Additionally, the computational cost of this tuning is strictly confined to the training phase. We believe the significant downstream benefits, specifically the ability to achieve higher expressivity, prune unimportant inputs, recover symbolic equations, and attain faster inference, fully justify the one-time investment in hyperparameter search.
> > >
> > > $\textbf{W4.}$ Lack of synthetic experiments for interpretability validation. Synthetic experiments with known ground truth would enable a clearer quantitative assessment of interpretability, but such evaluation is currently missing.
> > >
> > > $\textbf{Response to W4.}$ We appreciate the suggestion. Although we do not include fully synthetic experiments in this work, we provide multiple lines of evidence to validate the intepretability and input ranking of the PNAM. First, in the knot experiment (Table 2), the PNAM identifies the meridional translation (real part) and longitudinal translation as the two most important features, consistent with the findings of https://www.nature.com/articles/s41586-021-04086-x.pdf and https://arxiv.org/pdf/2404.19756. This agreement with domain-specific, independently validated findings supports the ability of the PNAM to recover scientifically relevant structure. Second, to offer visual and intuitive insight into the interpretability of the PNAM, we have added a high-dimensional image classification experiment using the MNIST data set in Appendix A.5. In this experiment, the PNAM achieves a test accuracy of 94.0\% using only $\sim$25\% of the pixels (with $M = 64$), demonstrating that the selected inputs are highly informative. Moreover, visual inspection confirms that the pruned pixel patterns align with human perceptual intuition: the remaining pixels clearly preserve recognizable features of the handwritten digits. Lastly, Fig. A.11 shows that the PNAM yields a consistent feature importance ranking across different values of $M$, indicating robustness of the interpretability mechanism.
> > >
> > > $\textbf{W5.}$ Limited evaluation of pruning and symbolic regression. Given the performance gap between the symbolic expression and the original MLP, it would strengthen the contribution to clearly justify why the derived symbolic form can still be regarded as interpretable and reliable.
> > >
> > > $\textbf{Response to W5.}$ We wish to clarify that the performance gap is not universal and that the drop is often related to a highly favorable compression ratio. In the phase field experiment (Section 3.2), the derived symbolic form in Table 5 actually achieves a lower test MSE ($2.25 \times 10^{-5}$) than fully connected MLPs in Table 4 ($7.05 \times 10^{-5}$). This demonstrates that the symbolic form is not just ``reliable,'' but likely generalizes better by filtering out noise and reducing overfitting, a common advantage of SR in physics. For the knot experiment, while there is a drop from $\sim$95\% (MLP) to $\sim$81\% (symbolic PNAM), we emphasize that we are compressing a model with tens of thousands of parameters into an explicit formula with fewer than 100 parameters. A test accuracy of 81\% for such a compact expression is highly non-trivial and competitive with other baselines (e.g., KANs in Table 3). We define interpretability here as the capability to derive explicit analytical laws with limited complexity. Even if a symbolic expression has slightly lower accuracy than a black box model, it offers a qualitative advantage: it reveals structure (e.g., the interaction between the meridional translation (real part) and longitudinal translation discovered in Table 2) that a black box model cannot, making it scientifically more valuable despite the numerical gap.

---

### Official Review · Reviewer_GGyN · 2025-11-05

**Soundness:** 2
**Presentation:** 3
**Contribution:** 2
**Rating:** 4
**Confidence:** 4

**Summary:**

This work attempts to provide a potential proof of the universal approximation property of PNAM through Theorem 2.1. Furthermore, by incorporating regularization constraints to enforce sparsity, as shown in Equation (9), the authors propose a possible approach to enhance the interpretability of PNAM.

**Strengths:**

1. It is helpful to provide a high-level overview or intuitive sketch before presenting the formal proof of Theorem 2.1.

2. The overall progression of the proof from Lemma A.2 to Lemma A.3 and then to Theorem A.1 is reasonable and clear, except for a few concerns I have raised (see Questions).

**Weaknesses:**

1. In the introduction, you should provide more details on how you define the interpretability of neural networks in your work. Does it refer to the significance of different input variables, the weighting parameters within the network, or another aspect?

2. In Section 2.1, which introduces the PNAM, more details should be provided about its underlying advantages and mechanisms. For example, why is a NAM needed beyond a standard neural network? How does it contribute to interpretability — by reducing the number of connections in a conventional NN, or in another way? In addition, how does PNAM enhance the expressiveness of NAM? Does the linear transformation from x to z play a beneficial role without compromising interpretability?

3. The mapping form of the PAM, as shown in Lemma A.1, appears to be more constrained than that of a conventional neural network, which could potentially reduce the hypothesis space for pattern recognition. How do you justify that the possible loss in expressiveness compared to a conventional NN is negligible or does not significantly affect performance? For example, can you justify why PNAM or NAM would not be more prone to underfitting compared to conventional neural networks? Otherwise, it is unclear why these new architectures are necessary, especially if their expressiveness is limited—even with a proof of universal approximation.

4. The notation of deg in Lemma A.2 should be explained to readers.

5. Even though the parameters may be non-unique, PNAM restricts the possible expressiveness to a subspace (as organized in Equation (A.3)) compared with a conventional neural network. How do you justify that the optimal expressiveness indeed lies within this subspace defined by PNAM?

**Questions:**

1. In Equation (1), $\epsilon$ represents the polynomial approximation error. Can it be reasonably treated as noise (for example, assumed to follow a Gaussian distribution)?

2. Regarding Equation (7), the proof of Theorem 2.1 does not appear rigorous. What justifies the assumption that the $\epsilon$ serving as an upper bound in Equation (7) is the same as the $\epsilon$ in Equation (5)? The essential condition for this relationship to hold is that the general form $F_2(X)$ in Equation (5) and the specific form of $F_2(X)$ in Equation (7)  is in the same continuous functional space.  Specifically, for conventional neural networks, this existence holds because they span the full polynomial space. However, in this work, you have constrained the representation to a subspace; therefore, how do you prove that the existence result still holds under this restriction?

3. Should interpretability help reduce uncertainty? However, with respect to Equation (9), it does not appear to guarantee this key property, since we can not guarantee the "significance" values are not wide in range. In that case, can the model still be considered interpretable?

4. What are the variables related to $\mathcal{L}_P$?

5. A key condition connecting Lemma A.2 and Lemma A.3 is the statement that ``Lemma A.3 can be proved by expanding $F(x_1, x_2)$, which produces the same set of monomials as the product of $f_1(x_1)$ and $f_2(x_2)$ in Lemma A.2.'' However, how do you justify that this condition always holds, or at least holds to some extent? Otherwise, certain forms of $F(x_1, x_2)$ may lack the necessary expressibility, making Theorem A.1 inapplicable.

---

> ### Author Response · Authors · 2025-11-24
> **Round 1, Part 1**
>
> We thank the reviewer for their critical evaluation and helpful insights. Please find below our responses to the comments and questions in Strengths, Weaknesses, and Questions.
>
> $\textbf{Strengths:}$
>
> $\textit{\textbf{S1.}}$ It is helpful to provide a high-level overview or intuitive sketch before presenting the formal proof of Theorem 2.1.
>
> $\textbf{Response to S1.}$ Thank you for your suggestion. We have rewritten Theorem 2.1 and its proof to clarify that the paragraph preceding Theorem 2.1 is, in fact, a high-level overview of the proof, with the first sentence corresponding to Eq. 6 and the next few sentences corresponding to Eqs. 7 and 8.
>
> We will respond to Q2 next, since it relates to Theorem 2.1.
>
> $\textit{\textbf{Q2.}}$ Regarding Equation (7), the proof of Theorem 2.1 does not appear rigorous. What justifies the assumption that the $\epsilon$ serving as an upper bound in Equation (7) is the same as the $\epsilon$ in Equation (5)? The essential condition for this relationship to hold is that the general form $F_2(X)$ in Equation (5) and the specific form of $F_2(X)$ in Equation (7) is in the same continuous functional space. Specifically, for conventional neural networks, this existence holds because they span the full polynomial space. However, in this work, you have constrained the representation to a subspace; therefore, how do you prove that the existence result still holds under this restriction?
>
> $\textbf{Response to Q2.}$ Please see the updated Theorem 2.1 and Eqs. 5--8. By the Stone--Weierstrass theorem, we can make the error in Eq. 6 arbitrarily small. By the universal approximation theorem of neural networks, we can also make the error in Eq. 8 arbitrarily small. If we make the errors in Eqs. 6 and 8 both smaller than $\epsilon / 2$, the final error in Eq. 5 is guaranteed to be smaller than $\epsilon$. Since Theorem A.1 proves that Eq. 7 is true, we can combine Eqs. 6--8 to show that Eq. 5 is true.
>
> $\textit{\textbf{S2.}}$ The overall progression of the proof from Lemma A.2 to Lemma A.3 and then to Theorem A.1 is reasonable and clear, except for a few concerns I have raised (see Questions).
>
> $\textbf{Response to S2.}$ Thank you for carefully reading our proofs in the Appendix. We will respond to Weaknesses (and questions similar in nature to the weaknesses) first before responding to (the remaining) Questions.

---

> ### Author Response · Authors · 2025-11-24
> **Round 1, Part 2**
>
> $\textbf{Weaknesses:}$
>
> $\textit{\textbf{W1.}}$ In the introduction, you should provide more details on how you define the interpretability of neural networks in your work. Does it refer to the significance of different input variables, the weighting parameters within the network, or another aspect?
>
> $\textbf{Response to W1.}$ This suggestion and question are very insightful. We have modified our Contribution in the Introduction to (1) motivate why the linear transformation is necessary and (2) differentiate between interpretability for the NAM and PNAM. We are currently working on a third numerical experiment using the MNIST data set (to be updated in the Appendix) to demonstrate the expressivity and dimensionality reduction capability of the PNAM for a high-dimensional problem, as well as visualize its interpretability/input pruning capability.
>
> $\textit{\textbf{W2.}}$ In Section 2.1, which introduces the PNAM, more details should be provided about its underlying advantages and mechanisms. For example, why is a NAM needed beyond a standard neural network? How does it contribute to interpretability — by reducing the number of connections in a conventional NN, or in another way? In addition, how does PNAM enhance the expressiveness of NAM? Does the linear transformation from x to z play a beneficial role without compromising interpretability?
>
> $\textbf{Response to W2.}$ Thank you. We have taken your suggestion into consideration. Please see the Remark in Section 2.1, which addresses the four questions.
>
> ``For the PNAM, each basis $g_i$ in Eq. 1 is now a function of a transformed variable $z_i$, as opposed to the original input $\chi_j$ for the NAM and other GAMs. The linear transformation $\boldsymbol{T}$, leading to universal approximation, enables the PNAM to capture interactions between the inputs, such as $\chi_1 \chi_2$ in Example A.1, that the NAM cannot. Nevertheless, for each data point, examining $\\{g_i\\}$ no longer tells us how the inputs $\boldsymbol{\chi}$ contribute to the prediction $\widehat{y}$. Instead, we describe in Section 2.5 how one can examine $\boldsymbol{T}$, which is constant for all data points, to determine which inputs are important for predictive accuracy. This and other aspects of interpretability (e.g., dimensionality reduction and feature pruning) are entirely missing from fully connected MLPs.''

---

> ### Author Response · Authors · 2025-11-24
> **Round 1, Part 3**
>
> $\textit{\textbf{W3.}}$ The mapping form of the PAM, as shown in Lemma A.1, appears to be more constrained than that of a conventional neural network, which could potentially reduce the hypothesis space for pattern recognition. How do you justify that the possible loss in expressiveness compared to a conventional NN is negligible or does not significantly affect performance? For example, can you justify why PNAM or NAM would not be more prone to underfitting compared to conventional neural networks? Otherwise, it is unclear why these new architectures are necessary, especially if their expressiveness is limited—even with a proof of universal approximation.
>
> AND
>
> $\textit{\textbf{W5.}}$ Even though the parameters may be non-unique, PNAM restricts the possible expressiveness to a subspace (as organized in Equation (A.3)) compared with a conventional neural network. How do you justify that the optimal expressiveness indeed lies within this subspace defined by PNAM?
>
> $\textbf{Response to W3 and W5.}$ Thank you for providing us with the opportunity to clarify our writing. While the NAM and standard GAMs have limited expressivity, which constrains the hypothesis class for pattern recognition and other tasks that demand higher expressivity, the PNAM achieves universal approximation. This universal approximation property, a necessary condition for expressivity, guarantees the existence of a PNAM architecture that can converge to an arbitrary continuous function. In fact, overcoming the limited expressivity of the NAM and GAMs (due to the constraint of using independent functions of individual variables) is a major contribution of our paper.
>
> Note that the ``mapping form'' in Lemma A.1---the representation of a 2D monomial via powers of a linear transformation $(\chi_1 + c_i \chi_2)^{p + q}$---is a proof technique used to establish the polynomial reproducing property; it is not an architectural restriction on the learned model. The learnable PNAM hypothesis class we study is
>
> $\mathcal{H}_\mathrm{PNAM} = \left\\{\boldsymbol{\chi} \mapsto \sum\_{i = 1}^M g_i\left(\sum\_{j = 1}^N T\_{ij} \chi_j\right) \bigg| \ \boldsymbol{T} \in \mathbb{R}^{M \times N} \ (\mathrm{learnable}), \ g_i: \mathbb{R} \rightarrow \mathbb{R} \ (\mathrm{univariate})\right\\},$
>
> where $\\{g_i\\}$ can be chosen as 1D universal approximators (e.g., 1D MLPs). In Appendix A.1, we show this class reproduces any multivariate polynomial: Lemma A.1 handles monomials in 2D; Lemma A.3 extends to arbitrary bivariate polynomials; Theorem A.1 lifts to $N$ dimensions by induction. Since polynomials are dense in $\mathcal{C}(\mathfrak{D})$ for a compact $\mathfrak{D}$ (Stone--Weierstrass), allowing $M$ and the single-variable bases to be sufficiently rich makes $\mathcal{H}_\mathrm{PNAM}$ dense in $\mathcal{C}(\mathfrak{D})$ (Theorem 2.1). Thus, an equation like Eq. A.3 in the Appendix is a representation result that exists for any polynomial once $\boldsymbol{T}$ and $\\{g_i\\}$ are chosen, not a restriction to a subspace.
>
> In practice, the expressivity of the PNAM is controlled by $M$ and the width/depth of the 1D bases, analogous to width/depth in conventional MLPs. Furthermore, the sparsity/rotation regularization constraints introduced for better interpretability are tunable. When set to zero, they leave the universal approximation result and capability unchanged. Meanwhile, the NAM and GAMs are special cases of the PNAM, in which the projection is simply a square identity matrix. However, universal approximation is lost for these cases, as the polynomial reproducing property cannot be guaranteed.

---

> ### Author Response · Authors · 2025-11-24
> **Round 1, Part 4**
>
> We will respond to Q5 next, since it relates to understanding the universal approximation property of the PNAM.
>
> $\textit{\textbf{Q5.}}$ A key condition connecting Lemma A.2 and Lemma A.3 is the statement that ``Lemma A.3 can be proved by expanding $F(x_1, x_2)$, which produces the same set of monomials as the product of $f_1(x_1)$ and $f_2(x_2)$ in Lemma A.2.'' However, how do you justify that this condition always holds, or at least holds to some extent? Otherwise, certain forms of $F(x_1, x_2)$ may lack the necessary expressibility, making Theorem A.1 inapplicable.
>
> $\textbf{Response to Q5.}$ Thank you for providing us with the opportunity to clarify our proof. Our statement is not setting conditions to make $\mathcal{F}(\chi_1, \chi_2)$ in the form of products in Lemma A.2, but rather the fact that $\mathcal{F}(\chi_1, \chi_2)$ can be rewritten as a linear combination of the monomials $\\{\chi_1^p \chi_2^q\\}_{p + q \leq \deg(\mathcal{F})}$. The latter are also the same monomials making up $f_1(\chi_1) f_2(\chi_2)$ in Lemma A.2 with $\deg(\mathcal{F}) = \deg(f_1) + \deg(f_2)$. Thus, the same proof works for $\mathcal{F}(\chi_1, \chi_2)$. Although we may have different right-hand side vectors for the linear systems of equations corresponding to the two different cases, we actually have the same (invertible) coefficient matrix. As a result, the same arguments can be applied, and the same conclusion holds.
>
> For clarification, we will elaborate on the proof of Lemma A.3, which encompasses Lemmas A.1 and A.2. In two dimensions, any polynomial $\mathcal{F}(\chi_1, \chi_2)$ admits the monomial expansion
> $$\mathcal{F}(\chi_1, \chi_2) = \sum_{p + q \leq \deg(\mathcal{F})} a_{pq} \chi_1^p \chi_2^q.$$
> For each fixed degree $r = p + q$, representing $\chi_1^p \chi_2^q$ as a linear combination of the powers $(\chi_1 + c_i \chi_2)^r$ yields a Vandermonde linear system whose coefficient matrix depends only on the chosen $\\{c_i\\}\_{i = 1}^{r + 1}$, i.e.,
> $$\chi_1^p \chi_2^q = \sum_{i = 1}^{r + 1} \zeta_{pqi} (\chi_1 + c_i \chi_2)^r,$$
> and is therefore the same across all pairs $(p, q)$ with $p + q = r$. The right-hand side encodes the target coefficients and hence changes with $\mathcal{F}$, but the matrix is invertible for distinct $\\{c_i\\}$. Thus, each degree block has a unique solution, and stacking across $r = 0, \dots, \deg(\mathcal{F})$ gives the desired representation of $\mathcal{F}$. Lemma A.2 (product case) can be viewed as the immediate step to show that a single linear map $\boldsymbol{T}$ of size $M = \deg(F) + 1$ suffices uniformly across degree blocks. With the 2D case established, the $N$-dimensional result follows by the stated induction in Theorem A.1.
>
> $\textit{\textbf{W4.}}$ The notation of deg in Lemma A.2 should be explained to readers.
>
> $\textbf{Response to W4.}$ The following line has been added below Eq. A.2: ``where $\\{\deg(f_i)\\}_{i = 1}^2$ denote the degrees of the single-variable polynomials $f_1$ and $f_2$, the highest exponents of the variables $\chi_1$ and $\chi_2$ in $f_1$ and $f_2$, respectively, with nonzero coefficients.''

---

> ### Author Response · Authors · 2025-11-24
> **Round 1, Part 5**
>
> $\textbf{Questions:}$
>
> $\textit{\textbf{Q1.}}$ In Equation (1), $\epsilon$ represents the polynomial approximation error. Can it be reasonably treated as noise (for example, assumed to follow a Gaussian distribution)?
>
> $\textbf{Response to Q1.}$ Yes, $\epsilon$ can be reasonably modeled as noise, and in many cases, it is appropriate to assume that it follows a Gaussian distribution (see, e.g., Eq. 3 of https://arxiv.org/pdf/2202.12482, which considers the (sparse) NAM). The term $\epsilon$ captures both approximation error arising from architectural choice and potential noise in the data. While the exact nature of $\epsilon$ may vary, this broad interpretation does not affect the validity of the theoretical result in Theorem 2.1.
>
> $\textit{\textbf{Q3.}}$ Should interpretability help reduce uncertainty? However, with respect to Equation (9), it does not appear to guarantee this key property, since we can not guarantee the ``significance'' values are not wide in range. In that case, can the model still be considered interpretable?
>
> $\textbf{Response to Q3.}$ Interpretability in our framework does not rely on enforcing a specific range for the coefficient values. Instead, the goal is to promote sparsity by driving coefficients associated with unimportant inputs and feature functions toward zero. To assess the stability and reliability of the feature importance, we perform multiple runs using both the same and different values of the weighting coefficients in Eq. 10, ultimately forming a consensus on which inputs are significant (see Appendix A.3). The validity of this input ranking is empirically validated through a pruning process: if pruning an input leads to a noticeable drop in predictive accuracy, that feature is added back into the model. This post hoc procedure does not require retraining, although retraining may increase accuracy at the expense of additional computational cost.
>
> Further support for the robustness of the ranking is provided in the newly added Figs. A.11 and A.12 in Appendix A.5 for the MNIST data set (with details of the experiment to be filled in), which demonstrate a consistent feature importance for different choices of $M$. While the magnitude of the significance values may vary, their relative ranking remains stable, allowing for meaningful interpretation. However, we emphasize that the PNAM is not a causal model, as interpretability here reflects predictive influence, not causal effect.
>
> $\textit{\textbf{Q4.}}$ What are the variables related to $\mathcal{L}_P$?
>
> $\textbf{Response to Q4.}$ It depends on the specific problems and constraints. We refer the reviewer to the original PINN paper (https://www.sciencedirect.com/science/article/pii/S0021999118307125), e.g., Eqs. 4 and 6, which differ depending on the problems and what physical constraints the users want to impose.
>
> Please consider raising your scores if we have adequately addressed your concerns.

---

### Official Review · Reviewer_tMah · 2025-11-11

**Soundness:** 3
**Presentation:** 3
**Contribution:** 2
**Rating:** 4
**Confidence:** 4

**Summary:**

The paper reviews Projected Neural Additive Models (PNAM) and targets to prove that this family can approximate any continuous target when given enough directions and flexible components. They also add practical tools for interpretability: encouraging the projection to be orthogonal and sparse, supporting monotonic/convex shape constraints on the one-dimensional components, and post-training symbolic compression of those components into compact expressions. Experiments on structured scientific problems demonstrate that PNAM outperforms a standard NAM and competes with other baselines at a similar capacity.

**Strengths:**

The math formulation and demonstration are easy to follow, also aligning with the claimed approximation capability.

The added practical interpretability tools (orthogonality/sparsity, shape constraints, symbolic compression) make it helpful to implement.

**Weaknesses:**

1. For Theorem 2.1, one clarification is needed: Is the universal approximation applied to an arbitrary continuous function?

2. From NAM to PNAM, it seems projection is what enables universal approximation here. If so, it may be clear and intuitive to show the mechanism, e.g., the projection mixes features and he 1-D parts then capture interactions that plain NAM cannot.

3. What’s the cost to get universality, like computation, memory, and sample complexity? As there is no free lunch.

4. While the claimed proof logic flow is easy to follow, the general idea needs some clarification. Why can this NN be universal at all? As formulated, the model abandons flexible raw-variable coupling, which is limiting. Is the point that the projection picks up the coupling among features into z, and then you keep separate 1-D MLP parameterizations on those directions?

5. Interpretability of NN, and NN for symbolic regression are mentioned as motivation. It's helpful to clarify where PNAM sits relative to other interpretable NN families like NALU (neural arithmetic units) and AI Feynman / SINDy / DSR, which are also stated in the paper. PNAM offers interpretable 1-D parts of a high-dimensional approximation, while NALU has exact function recovery?

**Questions:**

Please see Weakness.

---

> ### Author Response · Authors · 2025-11-24
> **Round 1, Part 1**
>
> We thank the reviewer for their thorough review and constructive suggestions. Please find below our responses to Weaknesses.
>
> $\textbf{Weaknesses:}$
>
> $\textbf{W1.}$ For Theorem 2.1, one clarification is needed: Is the universal approximation applied to an arbitrary continuous function?
>
> $\textbf{Response to W1.}$ Yes, the universal approximation property of the PNAM applies to an arbitrary continuous function ($\mathcal{F}_1$ in Theorem 2.1). We have revised Theorem 2.1 and the corresponding proof to clarify this point.
>
> We are currently working on a third numerical experiment using the MNIST data set (to be updated in the Appendix) as a high-dimensional problem, while also generating results to address the concerns in Weaknesses. We will respond again once these results are ready.

---

> > ### Author Response · Authors · 2025-11-27
> > **Round 1, Part 2**
> >
> > $\textbf{W2.}$ From NAM to PNAM, it seems projection is what enables universal approximation here. If so, it may be clear and intuitive to show the mechanism, e.g., the projection mixes features and the 1-D parts then capture interactions that plain NAM cannot.
> >
> > $\textbf{Response to W2.}$ Yes, the projection step is the key ingredient for universal approximation. While NAMs are restricted to axis-aligned additive forms $\sum g_i(\chi_i)$, PNAMs' projection $\boldsymbol{z} = \boldsymbol{T} \boldsymbol{\chi}$ allows the 1D nonlinearities to act on "rotated" or "mixed" coordinates. We have demonstrated this mechanism in Example A.1 of our paper. Here, we revisit the example of $\mathcal{F}(\chi_1, \chi_2) = \chi_1 \chi_2$, providing an alternative solution, and include one additional example. Note that both of these expressions are impossible for standard NAMs to reproduce.
> >
> > $\textbf{Example 1: Interactions (multiplication).}$
> > Consider the function $\mathcal{F}(\chi_1, \chi_2) = \chi_1 \chi_2$. A standard NAM cannot represent $\chi_1 \chi_2$ because it lacks cross terms (i.e., the cross derivatives of $\sum g_i(\chi_i)$ are always zero). A PNAM can capture this interaction using the polarization identity:
> > $$\chi_1 \chi_2 = \frac{1}{4}((\chi_1 + \chi_2)^2 - (\chi_1 - \chi_2)^2).$$
> > Here, the projection $\boldsymbol{T}$ learns to mix features into $z_1 = \chi_1 + \chi_2$ and $z_2 = \chi_1 - \chi_2$, while the 1D networks learn the quadratic functions $g_1(z_1) = z_1^2 / 4$ and $g_2(z_2) = -z_2^2 / 4$.
> >
> > $\textbf{Example 2: Rotated functions (oblique structure).}$
> > Consider a wave propagating along a diagonal: $\mathcal{F}(\chi_1, \chi_2) = \sin(\chi_1 + \chi_2)$. A standard NAM is forced to approximate this function as $g_1(\chi_1) + g_2(\chi_2)$, which cannot capture the diagonal wavefronts exactly. A PNAM can easily capture this interaction by using a projection row of $[1, 1]$ (yielding $z_1 = \chi_1 + \chi_2$) and the corresponding 1D network to learn $g_1(z_1) = \sin(z_1)$.
> >
> > $\textbf{W3.}$ What’s the cost to get universality, like computation, memory, and sample complexity? As there is no free lunch.
> >
> > $\textbf{Response to W3.}$ We thank the reviewer for this question. We have updated footnote 6 to correctly estimate the number of parameters of PNAMs for high-dimensional problems. If the number of inputs $N$ is significantly larger than $W$, the number of neurons in the widest layer, then the projection matrix will contribute meaningfully to the parameter count. (This update does not change the parameter count for the knot and phase field problems.) We have also added a sentence in footnote 6 to mention the relationship between the number of parameters and memory requirements. Compared to an MLP that uses the same number of layers $L$ and $W$, the PNAM has approximately $M$ times more parameters. Considering that memory requirements scale linearly with the number of parameters, the PNAM requires $M$ times more memory and is thus slower to train than the MLP (see Tables A.3 and A.4). However, the PNAM, after conversion to symbolic expressions, can be faster for inference than the MLP, agreeing with the findings in https://www.sciencedirect.com/science/article/pii/S0045782525000647. In the near future, we plan to parallelize the evaluations of the 1D bases (since they are independent of each other), which should reduce training times and further accelerate inference.
> >
> > On the other hand, for high-dimensional problems with numerous inputs and complex interactions, PNAMs can outperform NAMs while using significantly fewer parameters. We refer the reviewer to the newly added MNIST experiment in Appendix A.5 (with details of the experiment to be filled in), which highlights the dimensionality reduction capability and provides visual insight into the input pruning mechanism of PNAMs. Their linear transformation allows us to project the 784 pixels/features of the MNIST images onto a smaller feature space. Simply using $M = 8$, the corresponding PNAM outperforms a standard NAM, which requires one basis for every feature and has approximately 100 times more parameters (see Table A.5). Although PNAMs perform similarly to MLPs (i.e., additive models can now achieve the same expressivity as fully connected neural networks), PNAMs provide insight into which features are important for predictive accuracy that MLPs cannot (see Fig. A.12).

---

> > > ### Author Response · Authors · 2025-11-27
> > > **Round 1, Part 3**
> > >
> > > $\textbf{W4.}$ While the claimed proof logic flow is easy to follow, the general idea needs some clarification. Why can this NN be universal at all? As formulated, the model abandons flexible raw-variable coupling, which is limiting. Is the point that the projection picks up the coupling among features into z, and then you keep separate 1-D MLP parameterizations on those directions?
> > >
> > > $\textbf{Response to W4.}$ The question regarding why PNAMs can be "universal at all" is difficult for us to grasp. Recall that by the Stone--Weierstrass theorem, polynomials are dense in the space of continuous functions, i.e., they can approximate any continuous function. Similar to Theorem 2.1 in our paper, Hornik et al. (1989) make use of the Stone--Weierstrass theorem to establish that MLPs can approximate any continuous function to an arbitrary degree of accuracy (see https://www.sciencedirect.com/science/article/pii/0893608089900208). By proving in Theorem A.1 that PNAMs can reproduce any (multi-variable) polynomial, we leverage the Stone--Weierstrass theorem in Theorem 2.1 to prove that PNAMs are universal approximators. This universal approximation capability, due to the linear transformation and 1D nonlinearities, enables PNAMs to close the performance gap between NAMs and MLPs (see, e.g., Table A.5 in our paper and also Fig. 9 in https://www.sciencedirect.com/science/article/pii/S0045782525000647).
> > >
> > > If what the reviewer means by PNAMs abandoning "flexible raw-variable coupling" is why not use something like a GA$^2$M (e.g., see Eq. 1 of https://arxiv.org/pdf/2205.14120) and higher-order coupling functions to handle problems with interactions between the input variables, we note that the number of feature functions quickly blows up for this strategy. On the other hand, if the reviewer means why not use two nonlinear mappings like KANs (e.g., see Eq. 2.1 in https://arxiv.org/pdf/2404.19756), which relies on the Kolmogorov--Arnold theorem to establish universal approximation, we note that our proof differs from the Kolmogorov--Arnold theorem. In Appendix A.1, we show that the first nonlinearity is not necessary. (The short answer to the reviewer's last question is yes.) Specifically, the projection layer $\boldsymbol{z} = \boldsymbol{T} \boldsymbol{\chi}$ picks up couplings among features by creating linear combinations of the raw variables (effectively rotating the coordinate system). The 1D MLPs then learn independent nonlinear mappings on these coupled directions. Please also see our response to W2, where we provide concrete examples of this mechanism, allowing PNAMs to capture interactions (like $\chi_1 \chi_2$) and oblique structures that standard NAMs cannot.

---

> ### Author Response · Authors · 2025-11-27
> **Round 1, Part 4**
>
> $\textbf{W5.}$ Interpretability of NN, and NN for symbolic regression are mentioned as motivation. It's helpful to clarify where PNAM sits relative to other interpretable NN families like NALU (neural arithmetic units) and AI Feynman / SINDy / DSR, which are also stated in the paper. PNAM offers interpretable 1-D parts of a high-dimensional approximation, while NALU has exact function recovery?
>
> $\textbf{Response to W5.}$ We appreciate the opportunity to clarify the positioning of the PNAM relative to NALU and symbolic regression (SR) methods like AI Feynman and DSR. While NALU and SR families target exact symbolic recovery or specific arithmetic extrapolation, the PNAM occupies a distinct niche as a universal approximator that strikes a balance between expressivity and interpretability. Regarding expressivity, we introduce a PNAM in the MNIST experiment that achieves a test accuracy of 98.1\% while compressing the 784 pixels/features to a feature dimension of 64. Regarding interpretability, the PNAM provides insight into which pixels are crucial for maintaining predictive accuracy and enables post hoc pruning of the features.
>
> Unlike NALU, which is designed with inductive biases specifically for arithmetic operations, PNAMs are proven to approximate arbitrary continuous functions (Theorem 2.1), making them more applicable to general scientific modeling tasks where the underlying laws may not be purely arithmetic. As discussed in our paper (Section 2.5), the PNAM acts as a bridge for decomposing high-dimensional problems into 1D functions, which are significantly easier to learn via SR (see Figs. A.3, A.7, and A.8) if desired by the users. Nevertheless, we emphasize here (and also in Related work) that SR is primarily employed in PNAMs for pruning to reduce the number of parameters and to gain insight into the interactions of the input variables. The additive nature of PNAMs and other additive models, such as KANs, prevents them from recovering compact equations and concisely reproducing operators like division (see, e.g., Fig. 3 of https://openreview.net/pdf?id=TqzNI4v9DT). On the other hand, classical SR algorithms, such as AI Feynman, are designed for recovering compact physical laws and perform poorly on problems without precise functional forms (e.g., compare Figs. 1 and 3 in https://arxiv.org/pdf/2107.14351).

---

### Author Response · Authors · 2025-12-02
**Global Response**

We gratefully acknowledge all reviewers for their thorough evaluation and constructive feedback. Their comments have helped us improve the presentation of our key contributions and strengthen the overall quality of our paper. We have prepared a revised version of this paper, with key changes highlighted in red. Specifically, we have updated the Introduction and added a Remark to better articulate the advantages and mechanisms of our method; revised Theorem 2.1 and its proof to enhance the exposition of universal approximation, included additional results on training times, inference times, and convergence behaviors for the original two experiments; and added a new high-dimensional experiment using the MNIST data set to further demonstrate the expressivity and interpretability of our approach. We thank the reviewers for providing us with this opportunity to respond and improve our work.

---

### Meta-Review · Area_Chair_qQch · 2026-01-03

**Summary:**

This paper introduces Projected Neural Additive Models (PNAM), an extension of Neural Additive Models (NAM) that achieves universal approximation. The paper proves the universal approximation property using Stone-Weierstrass and proposes regularization and post-processing techniques for interpretability.

The proposed method PNAM has two advantages: universality and interpretability. However, the interpretability claims were questioned by all reviewers, and the answer was not completely satisfactory. Moreover, all the reviewers gave an original score below the acceptance threshold. Therefore, my suggestion is to reject the paper.

**Reviewer Concerns:**

Most of the reviewers' questions and concerns were addressed reasonably well. For example:
- The nature of the universality claims. These were responded reasonably well. The only caveat is that they didn't mention in the response (but did mention in the paper) that the domains of the functions need to be compact to use the SW theorem. This is a standard assumption (and proof technique), but the response "Yes, the universal approximation property of the PNAM applies to an arbitrary continuous function" misses that point.
- Interpretability claims. All the reviewers questioned in one way or another how is this model "interpretable". The response in the rebuttal about this point is a bit unsatisfactory since it doesn't explain why/how it is PNAM interpretable in general. It mentions that the architecture promotes sparsity and that the results on the MNIST example identify the relevant pixels, but it is not obvious from the answers that PNAM it provides a general mechanism for interpretability, nor that the paper engages with the interpretability literature in a rigorous way.
- One reviewer mentioned that the theoretical claims were not sufficiently rigorous. The authors responded by modifying the proof of the main theorem.
- Experiments. One reviewer mentioned that the experimental setup is insufficient. The authors added an experiment and explained why the other requests were not necessary for the paper's goal.

**Reviewer Scores:**

All the reviewers gave an original score below the acceptance threshold. One of them, however, was very positive and indicated that they might raise the score after the responses.

---

### Decision · Program_Chairs · 2026-01-26

Reject